# Deep coverage whole genome sequences and plasma lipoprotein(a) in individuals of European and African ancestries

Seyedeh M. Zekavat et al.[#]

Lipoprotein(a), Lp(a), is a modified low-density lipoprotein particle that contains apolipo-protein(a), encoded by *LPA*, and is a highly heritable, causal risk factor for cardiovascular diseases that varies in concentrations across ancestries. Here, we use deep-coverage whole genome sequencing in 8392 individuals of European and African ancestry to discover and interpret both single-nucleotide variants and copy number (CN) variation associated with Lp(a). We observe that genetic determinants between Europeans and Africans have several unique determinants. The common variant rs12740374 associated with Lp(a) cholesterol is an eQTL for *SORT1* and independent of LDL cholesterol. Observed associations of aggregates of rare non-coding variants are largely explained by *LPA* structural variation, namely the *LPA* kringle IV 2 (KIV2)-CN. Finally, we find that *LPA* risk genotypes confer greater relative risk for incident atherosclerotic cardiovascular diseases compared to directly measured Lp(a), and are significantly associated with measures of subclinical atherosclerosis in African Americans.

L ipoprotein(a), Lp(a), is a circulating lipoprotein comprised of a modified low-density lipoprotein (LDL) particle covalently bonded to apolipoprotein(a), apo(a)[1–3]. The apo(a) protein contains an inactive protease domain, kringle V domain, and ten kringle IV domains, including an extremely polymorphic kringle IV 2 copy number (KIV2-CN)[3], a large region spanning 5.5 kb, which consists of a pair of exons repeating between 5 to over 40 times per chromosome[4]. Increased KIV2-CN results in increased apo(a) size, which is inversely associated with plasma Lp(a) levels due to altered protein folding, transport, and secretion[5]. Twin studies have suggested that Lp(a) is highly heritable, with up to 90% heritability in both African and European populations[6–10]. However, the most recent genome-wide association studies have only explained approximately half of the genetic heritability[11]. Epidemiologic studies and genetic analyses in European and Asian populations have causally linked Lp(a) concentrations with atherosclerotic cardiovascular disease, independent of other plasma lipids including LDL cholesterol[12–15]. As a result, Lp(a) has emerged as a promising therapeutic target for atherosclerotic cardiovascular diseases.

Plasma Lp(a) distributions vary significantly among ethnicities but these differences are not explained by known differential KIV2-CN distributions between the ethnicities and are posited to be related to primary sequence[16]. Additionally, studies suggest that apo(a) isoform and Lp(a) concentration may have differential effects on coronary heart disease (CHD) odds[14]; however, distinguishing isoform-independent genetic effects on Lp(a) has required separate genotyping strategies, typically qPCR[17], in addition to genotyping single-nucleotide polymorphisms (SNPs). Deep-coverage (>20×) whole genome sequencing (WGS) provides the opportunity to determine the full range of genomic variation that influences Lp(a) concentration and isoform size, across the allele frequency spectrum and variant type among diverse individuals.

Here, we use deep-coverage WGS in 2284 Estonians, 2690 Finnish individuals, and 3418 African Americans to ascertain SNPs and indels across the genome, and structural variants at *LPA*, including KIV2-CN. We perform: (1) structural variant association analyses; (2) common variant association; (3) rare variant association in coding and non-coding sequence; and (4) Mendelian randomization (MR) analyses. Our goals are threefold: (1) to understand the full spectrum of genetic variation influencing Lp(a) and Lp(a)-cholesterol (Lp(a)-C); (2) to compare genetic differences between Europeans and African Americans; and (3) to determine the phenotypic consequences of *LPA* variant classes on incident clinical events and subclinical measures (Fig. 1).

Through WGS, we observe that Lp(a) is substantially heritable in both Europeans and African Americans despite notable interethnic differences in circulating biomarker concentrations. Furthermore, we use WGS to directly genotype *LPA* structural variation, including KIV2-CN. Through common variant and rare variant analyses, we dissect the genetic architecture of Lp(a), finding novel genetic associations and identifying sources of inter-ethnic genetic differences. Finally, using a new imputation model to estimate KIV2-CN, we show that distinct *LPA* variant classes differentially influence clinical and subclinical atherosclerosis.

## Results

**WGS and baseline characteristics**. A total of 8392 participants underwent deep-coverage (mean attained 33 × coverage) WGS: 3418 African Americans from the Jackson Heart Study (JHS) as part of the NIH/NHLBI Trans-Omics for Precision Medicine (TOPMed) program, 2284 Europeans from the Estonian Biobank

(EST), and 2690 Europeans from the Finland FINRISK study (FIN) (Supplementary Fig. 1). FIN WGS and whole-exome sequences were used to impute into 27,344 Finnish array data for analyses. Following quality control (Supplementary Table 1), a total of 119.4 M SNPs and 7.2 M indels were discovered across EST WGS, JHS WGS, and FIN imputation datasets analyzed (Supplementary Figs. 2, 3, Supplementary Table 2).

We obtained both Lp(a) and Lp(a)-C where available. 4767 individuals from EST and JHS WGS with Lp(a)-C available and 9272 individuals from the JHS WGS and FIN imputation dataset with Lp(a) available were included in analyses requiring these phenotypes. Lp(a)-C values were quantified using the Vertical Autoprofile (VAP) method, which measures cholesterol concentration via densitometry[18,19]. Lp(a) values were quantified using two immunoassay-based methods sensitive to the entire mass of the Lp(a) particle. Median Lp(a) levels in JHS (median (IQR) 46 (24–79) mg/dL) were nearly ten times higher than in FIN (5 (2–10) mg/dL), while the Lp(a)-C distribution was similar between EST (7 (5-9) mg/dL) and JHS (7 (5–11) mg/dL) (Supplementary Table 3, Supplementary Fig. 4a, b). Finnish individuals have among the lowest Lp(a) concentrations across European populations[20]. This may explain why we observe a 10-fold difference between JHS and FIN Lp(a) concentrations versus the 2–3 fold differences previously observed between African and European populations[16]. Among JHS individuals with both Lp(a) and Lp(a)-C available, the concentrations between these phenotypes were moderately correlated (Spearman correlation ($R_s$) = 0.46, $P = 2.4 \times 10^{-143}$) (Supplementary Fig. 5).

**Structural variant discovery and imputation of KIV2-CN**. Structural variants, notably KIV2-CN, at *LPA* have been previously shown to influence apo(a) size and Lp(a) concentration[17]. From the WGS data, we used GenomeSTRiP[21] to identify and genotype nine structural variants at the *LPA* locus (Fig. 2a, Supplementary Table 4), all rare except the KIV2-CN repeat. We mapped the reported 6 KIV2 repeats present in the hg19 reference genome[22], finding that the KIV2-CN repeat occurs between positions chr6:161032565–161067901 with each repeat copy containing 5534–5546 base pairs and two coding exons (Supplementary Fig. 6a). The KIV2-CN (quantified as the sum of the KIV2 allelic copy number across both chromosomes) distribution is slightly different between African American (mean 38.5 (SD 7.4)) and European (mean 43.7 (SD 6.2)) ethnicities, ranging between 12.0–84.6 copies (Supplementary Fig. 6b, Supplementary Table 5). In earlier work, we validated Genome STRiP copy number estimates using ddPCR[23], which establishes general accuracy for the quantified absolute copy number. To evaluate the precision of our KIV2-CN estimates, we utilized 123 pairs of siblings from JHS that were confidently identical-by-descent at both *LPA* 1 Mb window haplotypes (genotype concordance >99%), and found a very strong and robust correlation between sibling pair KIV2 copy number estimates ($r^2 = 0.989$) (Supplementary Fig. 7a-d).

*LPA* locus variants, namely rs3798220 and rs10455872, have been previously associated with KIV2-CN[14,15]. In the FIN WGS, these two SNPs account for 12% of the variance of directly genotyped KIV2-CN. To improve KIV2-CN estimation from SNPs, we developed an imputation model using 2,215 FIN with WGS and applied it to impute KIV2-CN in the 27,344 FIN with array-derived genotypes. In the FIN WGS, we applied the least absolute shrinkage and selection operator (LASSO) across high-quality (imputation quality > 0.8) variants with minor allele frequency (MAF) > 0.1% available in the FIN imputation dataset in a 4MB window around *LPA*, which yielded a 61-variant model to impute KIV2-CN (Supplementaary Fig. 8a). To understand the

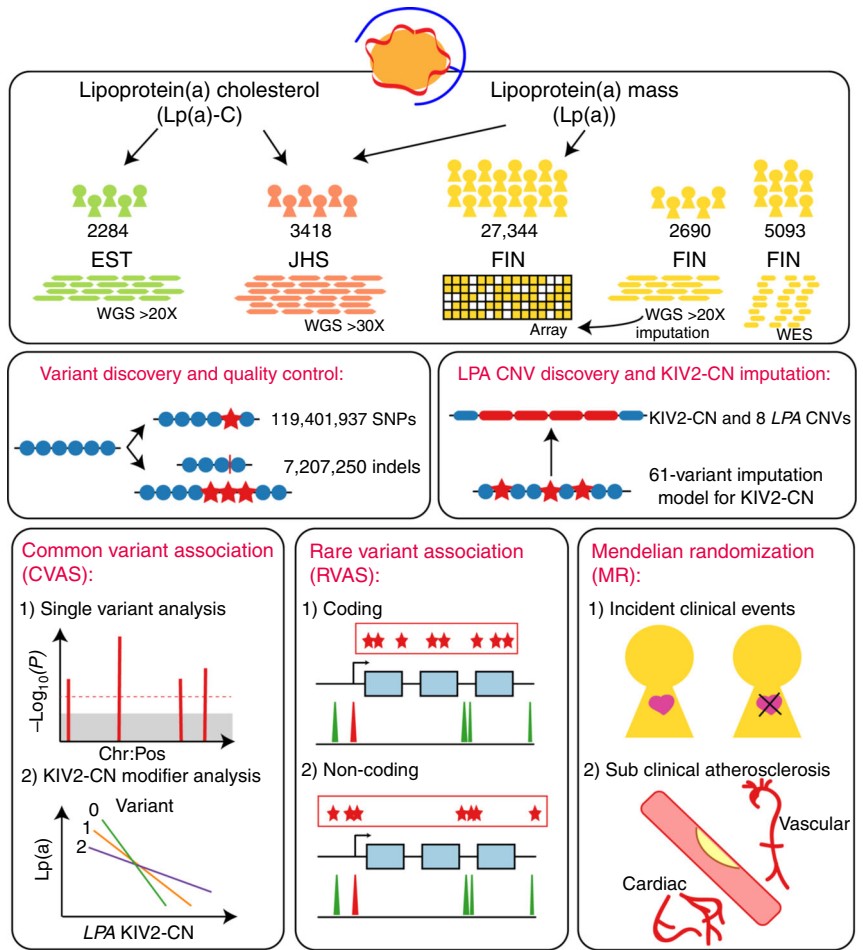

**Fig. 1** Schema of overall study design Analyses were stratified by phenotype, Lp(a) (mass) and Lp(a)-C, where available. Lp(a)-C analyses were performed using the following individuals with WGS data: 2284 individuals from the Estonian Biobank (EST) and 3418 individuals from Jackson Heart Study (JHS). Lp(a) mass analyses were performed using the same Jackson Heart Study participants, as well as array-derived genotypes from 27,344 Finnish FINRISK (FIN) individuals with imputation performed using 2690 FIN individuals with WGS and 5093 FIN individuals with WES. After quality control filters, 119,401,837 SNPs and 7,207,350 indels were discovered genome-wide across individuals analyzed. Structural variant discovery at the LPA locus was performed, finding KIV2-CN and eight additional rare CNVs. An imputation model was developed to impute KIV2-CN using 60 *LPA* locus variants. Three overarching analyses were subsequently performed: (1) Common variant analyses, (2) Rare variant analyses, and (3) Mendelian randomization. Among common and low-frequency variants with MAF > 0.1%, we performed single variant analysis, and separately, analyzed genetic modifiers of KIV2-CN's effect on Lp(a) and Lp(a)-C concentrations. We also performed rare variant analyses, aggregating rare variants (MAF < 1%) in (1) coding sequence and (2) putative functional non-coding sequence, and associated with Lp(a)-C. Lastly, we performed Mendelian randomization, using different classes of variants associated with Lp(a) as genetic instruments and associating these with incident clinical cardiovascular events in FIN and prevalent subclinical atherosclerosis in JHS, MESA, FHS, and OOA. CNV copy number variant, EST Estonian biobank, FHS Framingham heart study, FIN FINRISK, JHS Jackson Heart Study, KIV2-CN kringle IV-2 copy number, Lp(a) lipoprotein(a), Lp(a)-C lipoprotein(a) cholesterol, MAF minor allele frequency, MESA Multi-ethnic study of atherosclerosis, MR Mendelian randomization, OOA Old-Order Amish

relative importance of each of these 61 variants, a random forest model was applied (Fig. 2b, Supplementary Fig. 8b). Our model ascribed greatest importance to rs10455872, a previously described SNP associated with KIV2-CN[14,15]. The full 61-variant model in our validation dataset explained 60% of variation in genotyped KIV2-CN (Supplementary Data 1, Supplementary Fig. 6c, Fig. 2c). While low-frequency loss-of-function variants have been observed by us and others[24,25] within *LPA*, removal of these carriers did not significantly alter the relationship between KIV2-CN and Lp(a) across all individuals ($P = 0.48$).

We confirmed that both directly genotyped and imputed KIV2-CN were negatively associated with Lp(a)-C ($-0.05$ SD/CN, $P < 1 \times 10^{-61}$) and Lp(a) ($-0.07$ to $-0.08$ SD/CN, $P < 1 \times 10^{-190}$), across African American and European ethnicities (Fig. 3). KIV2-

CN alone explained 18% (Europeans) to 26% (African Americans) of variation in Lp(a), and for Lp(a)-C explained 14% of variation in both ethnicities. Introduction of 1/KIV2-CN to the multivariable model did not improve model fit for the relationship between KIV2-CN and Lp(a) ($P = 0.16$).

We sought to also determine whether combinations of summed KIV2-CN alleles equivalent to the same total had the same relationship with KIV2-CN. We observed that the relationship of homozygous KIV2-CN alleles (from 59 FIN individuals 95% homozygous-by-descent at the *LPA* locus) to Lp(a) was similar to the remaining association observed across all others ($P = 0.21$).

**Common variant associations**. To identify additional genomic variants associated with Lp(a) and Lp(a)-C, we performed

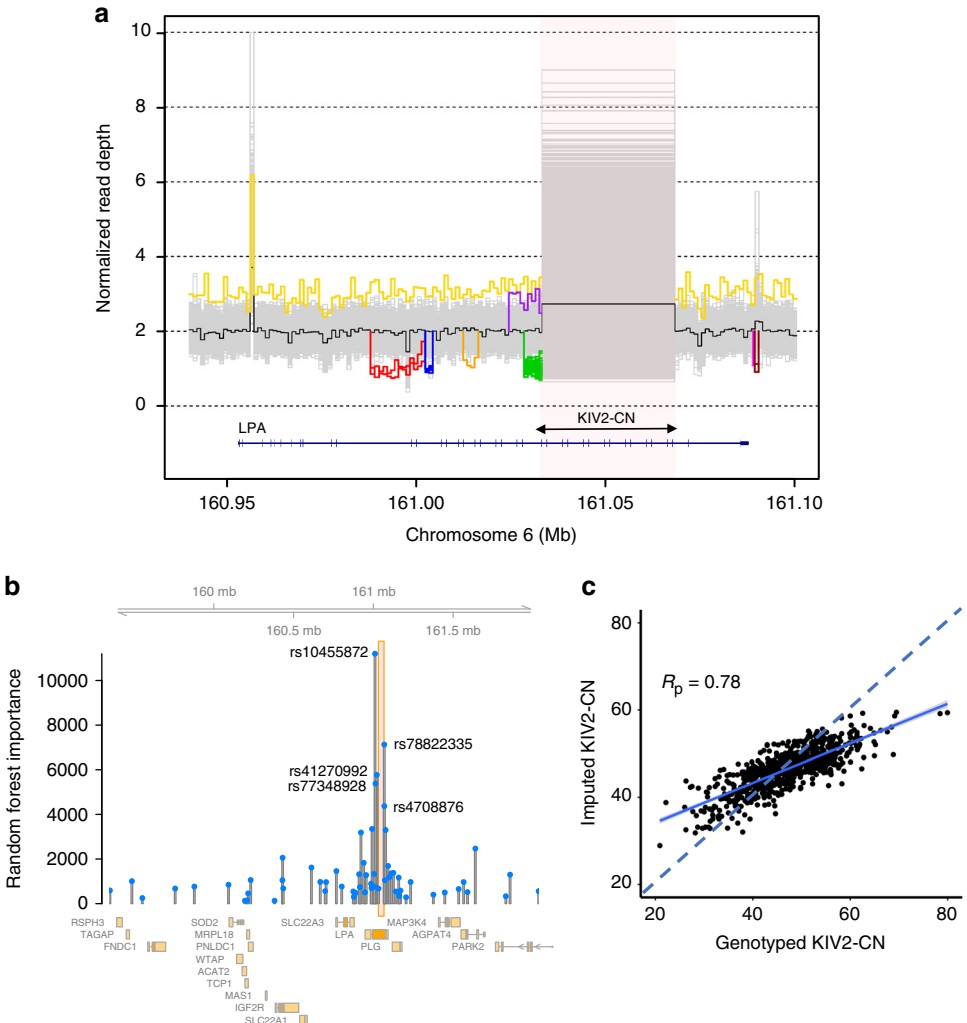

**Fig. 2** Structural variant discovery at the *LPA* locus and KIV2-CN imputation. **a** Nine separate copy number variants were discovered across the EST, JHS, and FIN whole genome sequences. Here, these are shown by plotting sample-level normalized read depth against the position along the hg19 reference genome at the *LPA* locus (with the black line denoting median read depth across all individuals). The KIV2-CN is shown in the highlighted region and each unique non-gray line outside of this region depicts a discovered structural variant (described further in Supplementary Table 4). **b** The random forest importance of each variant in the 61-variant KIV2-CN imputation model developed in FIN is shown against its genomic position, with KIV2-CN region highlighted and the top five rsIDs labeled. **c** Correlation of directly genotyped KIV2-CN and imputed KIV2-CN from 738 FIN individuals with WGS in the validation dataset (with Pearson correlation, $R_p = 0.78$). EST Estonian biobank, FIN FINRISK, JHS Jackson Heart Study, KIV2-CN kringle IV-2 copy number

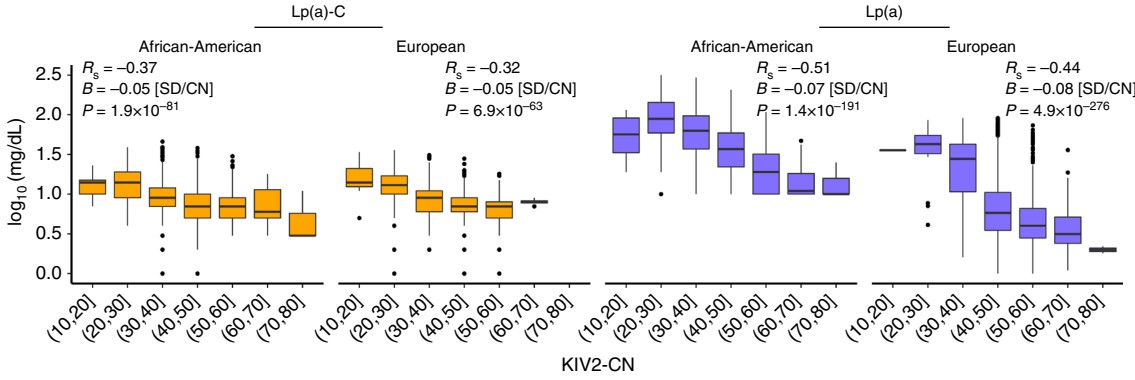

**Fig. 3** KIV2-CN association with Lp(a) phenotypes. Directly genotyped KIV2-CN (in EST and JHS) and imputed KIV2-CN (in FIN) are inversely associated with Lp(a) and Lp(a)-C. EST Estonian biobank, FIN FINRISK, JHS Jackson Heart Study, KIV2-CN kringle IV-2 copy number, Lp(a) lipoprotein(a), Lp(a)-C lipoprotein(a) cholesterol; $R_p$ = Pearson correlation; $R_S$ = Spearman correlation

genome-wide common variant (MAF > 0.1%) association analyses using a linear mixed model, conditioning on KIV2-CN. Association was performed at the cohort-level and followed by trans-ethnic meta-analysis. We analyzed a total of 32,695,476 variants for Lp(a)-C and 31,652,301 variants for Lp(a), identifying common variants at 3 loci at conventional genome-wide significance ($P < 5 \times 10^{-8}$) for Lp(a)-C at *LPA* (rs140570886, $P = 3.3 \times 10^{-30}$), *CETP* (rs247616, $P = 6.1 \times 10^{-10}$), and *SORT1* (rs12740374 $P = 1.0 \times 10^{-21}$), and 2 genome-wide significant loci for Lp(a) at *LPA* (rs6938647, $P = 4.7 \times 10^{-129}$), and *APOE* (rs7412, $P = 1.3 \times 10^{-23}$) (Supplementary Fig. 9-11; Supplementary Data 2, 3).

The lead *SORT1* locus variant, rs12740374, has been previously causally associated with LDL cholesterol[26]. Here, Lp(a)-C association for rs12740374 was not substantially altered conditioned on either LDL cholesterol (Fig. 4a) or apolipoprotein B (Supplementary Fig. 12). Common variants at *CETP* are associated with HDL cholesterol[27] and the lead *CETP* locus variant for Lp(a)-C, rs247616, is no longer significant after conditioning on HDL cholesterol (Supplementary Fig. 13). Lp(a)-C is strongly associated with HDL cholesterol ($B = 0.41$ SD Lp(a)-C/SD HDL, $P = 2.9 \times 10^{-191}$); notably, HDL and Lp(a) particles have similar densities potentially influencing Lp(a)-C

measurement accuracy[28]. Finally, rs7412 (*APOE* p. Arg176Cys), denoting the major APOE2 polymorphism, has been previously associated with LDL cholesterol[29] and recently with Lp(a) in a meta-analysis[11]. The association of rs7412 with Lp(a) is diminished when conditioning on LDL cholesterol but remains strongly associated (before conditioning: $B = -0.25$ SD, $P = 1 \times 10^{-23}$, after conditioning: $B = -0.18$ SD, $P = 5 \times 10^{-16}$) (Fig. 4b).

On average, *LPA* locus genetic variants yielding a 1 SD increase in Lp(a) yield a 0.48 SD increase in Lp(a)-C, similar to the observational correlation between the two phenotypes (Supplementary Fig. 14). Iterative conditional analyses at the *LPA* locus showed that, for Lp(a)-C there are 2 (JHS) and 3 (EST) independent genome-wide significant variants, (Supplementary Table 6a, b), while for Lp(a) there are 13 (JHS) and 30 (FIN) independent genome-wide significant variants (Supplementary Data 4) (Supplementary Fig. 15a, b), similar to the number of independent variants from past studies[11,17,30,31]. We replicated Lp(a) associations for two known *LPA* loss-of-function (LOF) alleles[24,25]: splice donor variant rs41272114 ($B = -0.7$ SD, $P = 8 \times 10^{-77}$) and splice acceptor variant rs143431368 ($B = -0.5$ SD, $P = 2 \times 10^{-26}$), and also discovered a novel LOF variant, a splice acceptor variant in exon 28 only observed African Americans in JHS: rs199583644 (MAF = 0.28%, $B = -1.5$ SD, $P = 3 \times 10^{-13}$).

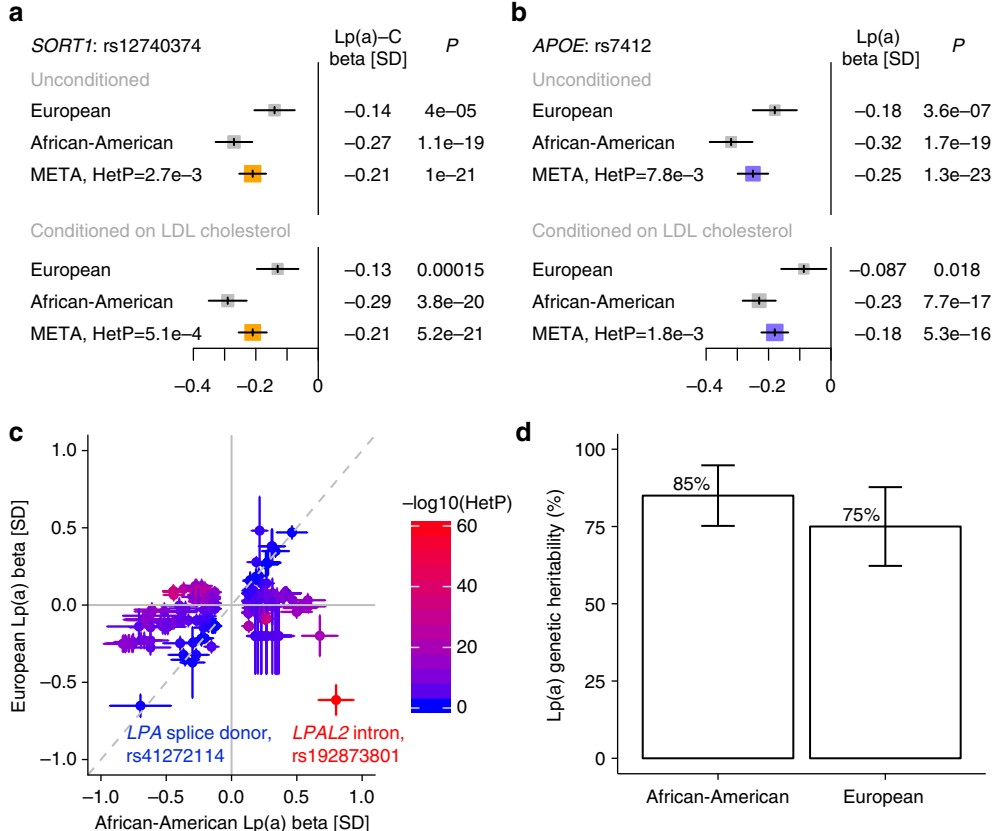

**Fig. 4** Trans-ethnic *LPA* and non-*LPA* loci associations with lipoprotein(a) phenotypes. In trans-ethnic meta-analysis of single variant results adjusted for KIV2-CN, we observed two associations ($P < 5 \times 10^{-8}$) at loci distinct from *LPA* and independent of other conventional lipid measures: *SORT1* for Lp(a)-C and *APOE* for Lp(a). **a–b** Associations (Betas in SD and 95% CI) for top variants at the *SORT1* and *APOE* loci are shown by ethnicity. The *SORT1* and *APOE* loci have been previously associated with LDL cholesterol. Thus, associations conditional on LDL cholesterol are also presented. The effect size for *SORT1* is preserved after conditioning on LDL cholesterol while the effect size for *APOE* is slightly reduced but still genome-wide significant. **c** Standardized effect estimates for variants at the *LPA* locus (*LPA* TSS ± 1 Mb) attaining $P < 5 \times 10^{-8}$ in JHS are shown comparing effects in JHS (African Americans) with FIN (European Americans). Color demonstrates inter-ethnic effect difference as measured by heterogeneity *P*. Similar effects are observed for a known null (splice donor) mutation in *LPA* but strongly diverging effects are observed for a distinct nearby *LPAL2* intronic variant. **d** Genetic heritability estimates using variants with MAF > 0.001 for normalized Lp(a) were acquired for African Americans in the whole-genome sequenced JHS cohort and for Europeans in the genotyped and imputed FIN cohort. Here, heritability and 95% CI are shown without adjusting for KIV2-CN. KIV2-CN kringle IV-2 copy number, HetP heterogeneity *P*, Lp(a) lipoprotein(a), Lp(a)-C lipoprotein(a) cholesterol, MAF minor allele frequency, TSS transcription start site

Next, we compared inter-ethnic effects of *LPA* locus variants attaining sub-threshold significance ($P < 1 \times 10^{-4}$) in either ethnicity for Lp(a) and Lp(a)-C. Spearman rank correlation of genetic effects between the two ethnicities for Lp(a)-C was 0.38 and for Lp(a) 0.16 (Supplementary Fig. 16a, b). Moderately associated ($P < 1 \times 10^{-2}$) *LPA* locus variants largely private in African Americans (FIN MAF < 0.1%) had larger absolute effects across MAFs compared to such variants observed in both ethnicities ($P = 3 \times 10^{-32}$) (Supplementary Fig. 17a, b). In comparing betas from genome-wide significant variants in African Americans with betas from the same variants in Europeans (Fig. 4c), we found the strongest inter-ethnic heterogeneity ($HetP = 9.8 \times 10^{-64}$) at an *LPAL2* intronic variant at the *LPA* locus (rs192873801, MAF 2.8% in JHS and 2.7% in FIN) with strongly divergent effects between the two ethnicities: +0.80 SD in JHS ($P = 3.8 \times 10^{-32}$) and −0.61 SD in FIN ($P = 2.0 \times 10^{-35}$) (Supplementary Fig. 18). We noted these variants to be on separate haplotypes for JHS and FIN (Supplementary Fig. 19). Notably, the *LPA* loss-of-function variant rs41272114, shows similarly strong effects in both ethnicities ($HetP > 0.05$).

Early family studies in Europeans and Africans have suggested the heritability of Lp(a) to be between 51% and 90%[6–10]. A recent array-based genotyping study in KORA estimated 49%[11] of variance in Lp(a) from genome-wide heritability analysis of 6,002 Europeans. From WGS, we now estimate genetic heritability in African Americans and Europeans, respectively, to be 85% (SE 5%) and 75% (SE 7%) for Lp(a), and 52% (SE 7%) and 75% (SE 34%) for Lp(a)-C (Fig. 4d).

**Common variant association and KIV2-CN modifier analyses.** To determine if there are variants that influence the relationship between KIV2-CN and Lp(a)-C or Lp(a) concentrations, we performed variant-by-KIV2-CN interaction analyses at a 4MB window around *LPA*. We identified three independent modifier variants at this locus which influenced the relationship between KIV2-CN and Lp(a)-C (rs13192132, $P = 1.73 \times 10^{-15}$, rs1810126, $P = 6.84 \times 10^{-14}$, rs1740445, $P = 6.35 \times 10^{-9}$) (Fig. 5) and were consistent across ethnicities (Supplementary Table 7, Supplementary Fig. 20a, b). Sensitivity analyses of interactions was performed to assess for confounding from 1) haplotype effects and 2) single variants tagged through LD[32,33]. All three variants show association with Lp(a)-C individually ($P < 0.05$), but are not correlated with KIV2-CN genotype (Pearson correlation $r^2 < 0.1$) (Supplementary Table 8). Furthermore, interaction associations persisted after conditioning on variants independently associated with Lp(a)-C (Supplementary Table 9).

Genomic context interrogation using adult liver regulatory annotations from the Roadmap Epigenome Project[34] showed that the top modifier variant in EST, a 3-base deletion, rs4063600 (TAGG > T, $B = + 0.03$ SD Lp(a)-C/CN/allele, $P = 2.96 \times 10^{-12}$), is in strong LD with rs13192132 ($r^2 = 0.88$) and overlies significant H3K4me3 and H3K27ac peaks ($P < 1 \times 10^{-2}$) 7,508 bases downstream of the *LPA* transcription start site (TSS) (Supplementary Fig. 21a). We additionally performed variant-by-KIV2-CN modifier analyses for Lp(a) using the JHS WGS (Supplementary Fig. 21b). A complete list of cohort-specific, LD-clumped significant variants are provided in Supplementary Data 5.

**Rare variant analysis by coding and non-coding burden tests.** Rare and low-frequency disruptive coding variants within *LPA* have been previously associated with Lp(a)[24,25]. Here, we performed two coding rare variant analyses studies (RVAS) aggregating rare (MAF < 1%) variants which were (1) LOF or missense deleterious by in silico prediction tools[35], or (2) non-

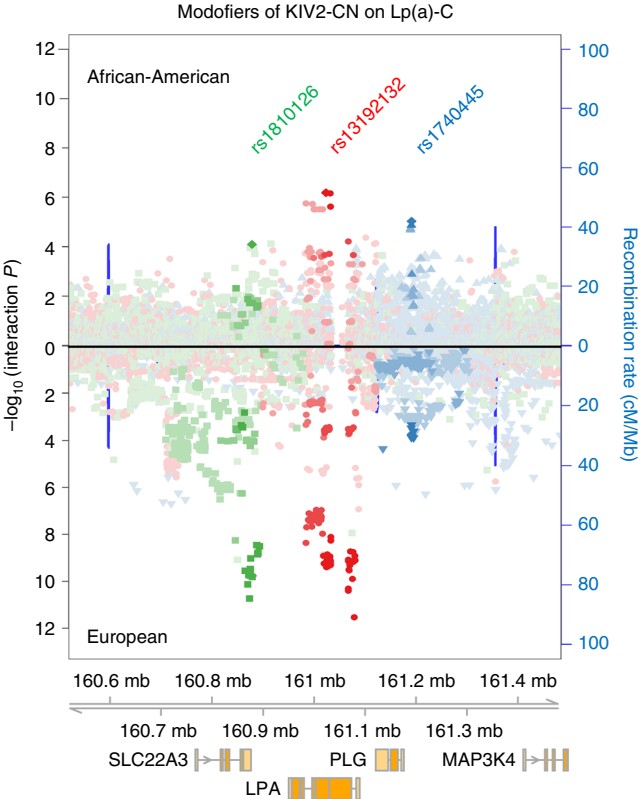

**Fig. 5** Genetic modifiers of KIV2-CN's effect on lipoprotein(a) cholesterol. Three independent genetic modifiers of KIV2-CN's effect on Lp(a)-C were discovered at the *LPA* locus. Regional association plots showing the variant-by-KIV2-CN interaction $P$ values of all variants within a 1 Mb window of the *LPA* TSS are shown for African Americans (top) and Europeans (bottom), highlighting variants in linkage disequilibrium with rs1810126 (green), rs13192132 (red), and rs1840445 (blue), the top independent genome-wide significant variants (interaction $P < 5 \times 10^{-8}$) upon meta-analysis. KIV2-CN kringle IV-2 copy number, Lp(a)-C lipoprotein(a) cholesterol, TSS transcription start site

synonymous, within their respective genes, and performed association with Lp(a)-C, adjusting for KIV2-CN. All analyses were done separately for JHS and EST and meta-analyzed. While no genes reached significance in either analysis after accounting for multiple-hypothesis testing, we observed suggestive evidence for *LPA* in both coding RVAS tests ($P = 7 \times 10^{-4}$ for LOF and missense deleterious mutations, $1 \times 10^{-4}$ for non-synonymous mutations) (Supplementary Data 6, 7, Supplementary Fig. 22a, b).

We also interrogated whether there was evidence of rare, non-coding variants aggregated within regulatory sequences uniquely detected by WGS that influence Lp(a)-C. We performed three non-coding RVAS using the variant groupings described in the Methods along with Roadmap epigenome data[34] from adult liver, the main tissue where *LPA* is expressed (Supplementary Fig. 23, Supplementary Fig. 24). The only genome-wide significant association was for an intron of *SLC22A3* at 6:160851000-160854000 with Lp(a)-C ($P = 4.5 \times 10^{-8}$) (Supplementary Data 8-13). Similarly, rare variants in a putative regulatory domain of *SLC22A3* were recently shown to be associated with Lp(a) in a sliding window analysis using low-coverage whole genomes[36]. However, we found that conditioning on *LPA*'s KIV2-CN, 128 kb away, mitigated the observed association ($P = 4.3 \times 10^{-3}$, Supplementary Data 8, 9). Upon conditioning on KIV2-CN, while no sliding windows reached statistical significance, the top window was 6:160,939,500–160,942,500 ($P = 1.6 \times 10^{-4}$), 13 kb

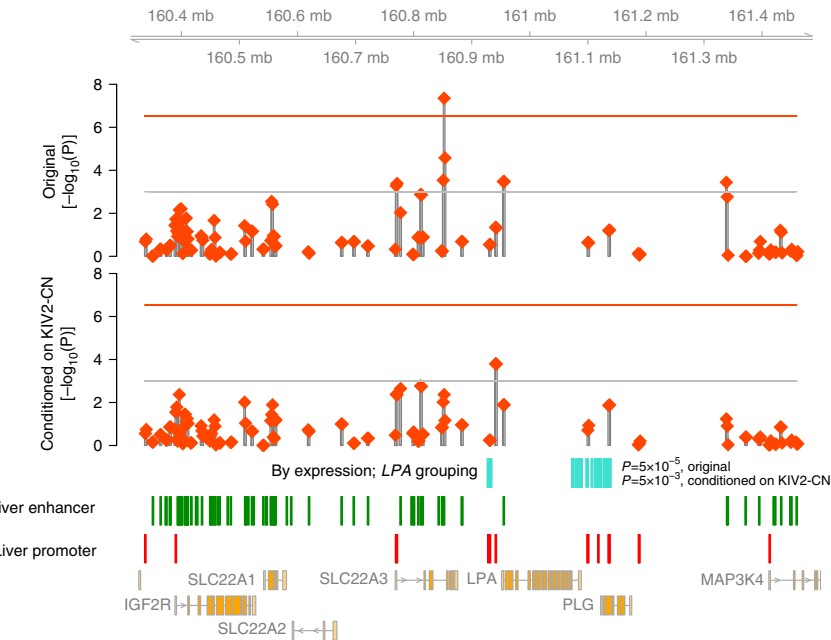

**Fig. 6** Rare variant non-coding burden analyses. A schematic of rare variant association results from (1) aggregating rare variants in adult liver enhancers or promoters and strong DHS ($P < 10^{-10}$) within 3 kb sliding windows, and (2) aggregating rare variants in liver enhancers grouped to *LPA* via the "By expression" in silico prediction method. In the top two panels, each red diamond represents the meta-analyzed mixed-model SKAT P-value with Lp(a)-C of rare (MAF < 1%), non-coding variants overlapping liver enhancer or promoter annotations in strong DHS (P(DHS) < 1e-10) grouped in a 3 kb window, before adjusting for KIV2-CN (top, "Original") and after adjusting for KIV2-CN (bottom, "Conditioned on KIV2-CN"). The horizontal red lines denote the genome-wide Bonferroni significance threshold given the number of unique windows analyzed. The horizontal gray lines denote the Bonferroni significance threshold within this 1MB region around *LPA*. The regions incorporated into the "By Expression" grouping to *LPA* are shown in aqua, along with the respective associations of rare non-coding variants in these regions before and after conditioning on KIV2-CN. Annotated adult liver enhancers (green bars) and promoters (red bars) overlapping strong DHS are included above protein-coding genes from Ensembl. DHS DNAse hypersensitivity sites, Lp(a)-C lipoprotein(a) cholesterol, MAF minor allele frequency, SKAT Sequence Kernal Association Test

downstream of the *LPA* transcription end site and overlapping three annotated ORegAnno[37] CTCF binding sites (Fig. 6).

Interrogation of rare enhancer variants predicted to influence *LPA* expression in liver[38] showed nominal evidence of association with Lp(a)-C before ($P = 5 \times 10^{-5}$) and after ($P = 1 \times 10^{-3}$) conditioning on KIV2-CN (Fig. 6, Supplementary Fig. 25). However, other putative gene-linked rare enhancer variants at the *LPA* locus, including the aforementioned *SLC22A3* (Supplementary Fig. 26), also demonstrate nominal associations, highlighting current challenges in both mapping associated regulatory elements to causal genes through in silico approaches and discerning the relative impacts of potentially pleiotropic regulatory elements.

**Mendelian randomization.** Genetic variation at the *LPA* locus is an optimal instrument for MR as it strongly and specifically influences circulating Lp(a) levels. Past studies have performed Lp(a) MR across clinical and metabolic traits using genetic risk scores comprised of between 1–18 variants[14,39,40]. Here, we performed MR using three different genetic instruments per cohort to distinguish variant classes influencing Lp(a) phenotypes: (1) an expanded genetic risk score, "GRS," comprised of the sum of the KIV2-CN-adjusted variant effects from LD-pruned variants in a ~4MB window around *LPA* with sub-threshold significance ($P < 1 \times 10^{-4}$); (2) a "KIV2-CN" score using the directly genotyped or imputed KIV2-CN; and (3) a combined "GRS + KIV2-CN" score combining scores from (1) and (2). Each genetic instrument was normalized such that 1 unit increase in the score was equal to 1 SD increase in Lp(a) (or Lp(a)-C). In African Americans, 235 variants were used towards the Lp(a) GRS and 39 towards the Lp(a)-C GRS (Supplementary Data 14).

In Europeans, 399 variants were used towards the Lp(a) GRS and 49 towards the Lp(a)-C GRS (Supplementary Data 14). The GRS + KIV2-CN score explains 45–49% of Lp(a) variance and 20% of Lp(a)-C variance (Supplementary Fig 27, Supplementary Table 10).

Association of GRS + KIV2-CN with 10 incident clinical phenotypes from the FIN imputation dataset ($N = 27,344$) (Fig. 7a, Supplementary Table 11) demonstrated anticipated associations for incident cardiovascular diseases (HR 1.18/Lp(a) SD, $P = 1 \times 10^{-5}$), comprising incident myocardial infarction (HR 1.23/Lp(a) SD, $P = 8 \times 10^{-4}$), CHD (HR 1.25/Lp(a) SD, $P = 7 \times 10^{-7}$), and stroke (HR 1.27/Lp(a) SD, $P = 1 \times 10^{-3}$). For given effect on Lp(a), the GRS had a larger effect on incident CHD risk (HR 1.36/Lp(a) SD, $P = 7.6 \times 10^{-8}$) than KIV2-CN (HR 1.03/Lp(a) SD, $P = 0.17$). Similar trends were observed for incident myocardial infarction. While the KIV2-CN score alone was not as strongly associated with cardiovascular outcomes ($P > 0.05$), its estimated effect with incident MI (HR = 1.16) was similar to recent estimations in a MI case-control analysis[14]. Thus, power for MR using the KIV2-CN instrument may be hindered due to a limited number of incident MI cases and modest effect conferred by KIV2-CN. These results suggest that knowledge of *LPA* variant class genotypes may provide additional information on cardiovascular risk beyond circulating Lp(a) levels.

To determine whether *LPA* genomic variants influence the accumulation of subclinical cardiovascular atherosclerosis, we associated both the Lp(a) and Lp(a)-C genetic instruments with computed tomography-derived measures of atherosclerosis in the coronary arteries (CAC) and abdominal aorta (AAC) in 3221 of African ancestry and 3361 of European ancestry (Supplementary

 

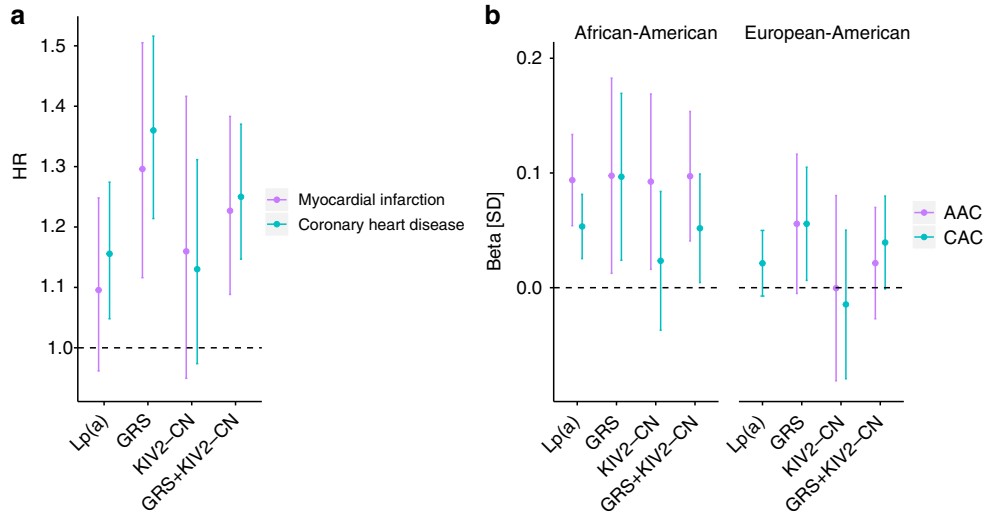

**Fig. 7** Association of *LPA* variant classes with atherosclerosis. Mendelian randomization was performed using three genetic instruments: a weighted genetic risk score using variants conditioned on KIV2-CN at a 4 Mb window around *LPA* (GRS), a KIV2-CN score, and a combined GRS + KIV2-CN score, and compared to the observational effects. The genetic instruments were all normalized such that 1 unit increase in the score is equal to 1 SD increase in Lp (a). **a** Associations (HR and 95% CI) of incident coronary heart disease (1056 cases; 21,207 controls) and myocardial infarction (580 cases; 21,377 controls) with the Lp(a) measurement and with genetic instruments among the genotyped and imputed FIN individuals (exact values in Supplementary Table 30). **b** Associations (Beta and 95% CI) of Lp(a) measurements and respective genetic instruments with standardized markers of subclinical atherosclerosis (CAC and AAC) among whole genome sequences of African Americans from 1701 JHS and 932 MESA participants, as well as European Americans from 1536 FHS and 1651 MESA participants (Supplementary Table 12). These data indicate that (1) a comprehensive Lp(a) genetic instrument (GRS + KIV2-CN) provides improved risk assessment compared to the Lp(a) phenotype, and (2) further stratifying this comprehensive instrument into separate Lp(a) variant classes provides additional risk stratification in that genomic sequence variants independent of KIV2-CN (i.e., GRS) have a stronger influence on clinical atherosclerosis compared to KIV2-CN. AAC Abdominal aortic calcium, CAC coronary artery calcium, CI confidence interval, FHS Framingham Heart Study, FIN FINRISK, GRS genetic risk score, HR hazard ratio, JHS Jackson Heart Study, KIV2-CN kringle IV-2 copy number, Lp(a) lipoprotein(a), Lp(a)-C lipoprotein(a) cholesterol, MESA Multi-Ethnic Study of Atherosclerosis

Table 12, Fig. 7b, Supplementary Fig. 28). Among African Americans without prevalent clinical atherosclerotic cardiovascular disease, the comprehensive (GRS + KIV2-CN) genetic instruments for both Lp(a) and Lp(a)-C demonstrated association with subclinical atherosclerosis in two vascular locations (coronary arteries and abdominal aorta): Lp(a) (AAC: $B = 0.97$, $P = 7.38 \times 10^{-4}$; CAC: $B = 0.052$, $P = 0.032$), and Lp(a)-C (AAC: $B = 0.123$, $P = 6.3 \times 10^{-3}$; CAC: $B = 0.074$, $P = 0.039$). Notably, this is the first known demonstration of Lp(a) or *LPA* genomic variants affecting atherosclerotic risk in African Americans. A prior study of African Americans from the Dallas Heart Study found no association between Lp(a) phenotype and subclinical measures of atherosclerosis, such as CAC[41]. With a larger sample size and use of a genetic instrument, our study has greater power for detecting this association among African Americans. Associations were less pronounced for European Americans between both observational and genetic instruments and subclinical atherosclerosis. The strongest association for European Americans was with Lp(a) GRS independent of KIV2-CN (CAC: $B = 0.056$, $P = 0.027$).

## Discussion

We characterized the genetic architecture of Lp(a) and Lp(a)-C using deep-coverage WGS in 8,392 Europeans and African Americans across allele frequencies and classes. While we observe that Lp(a) is highly heritable in Europeans and African Americans, distinct and common genetic determinants influence concentrations. Using a comprehensive genetic instrument that separately imputes apo(a) isoform, we show that knowledge of *LPA* genotypes can better inform incident cardiovascular disease risk prediction than just knowledge of Lp(a) biomarker level.

These observations permit several conclusions. First, through whole-genome sequencing and imputation, we observe substantial genetic heritability of Lp(a)—85% (SE 5%) in African Americans and 75% (SE 6%) in Europeans. We leverage this observation to systematically dissect the heritable components of Lp(a) across the two ethnicities. Through single variant analysis, we find a novel locus for Lp(a)-C, *SORT1*, whereby the top variant (rs12740374) reduces plasma Lp(a)-C concentrations in both ethnicities and is independent of LDL cholesterol levels, thereby providing evidence for the sortilin receptor as a novel component in Lp(a)-C metabolism. Through genetic modifier analysis, we find evidence of three loci which affect the relationship between KIV2-CN and Lp(a)-C similarly across both ethnicities. We replicate evidence supporting rare coding variation at *LPA* influencing Lp(a); however, observed associations of aggregates of rare non-coding variation appeared to be largely explained by *LPA* structural variation, namely KIV2-CN.

Second, we observed high heritability in diverse ethnicities despite notable inter-ethnic differences in circulating biomarker concentrations. Upon finding that similar Lp(a) effect sizes are conferred per KIV2 copy in African Americans and Europeans, we delved further into KIV2-independent effects conferred by variants at the *LPA* locus. Among distinct sequence variation, we notably observed an *LPAL2* intronic variant with significant yet opposing effects in each ethnicity, likely indicating influences from haplotype structure or gene-environment interactions. Altogether, *LPA* locus variants largely private to African Americans (FIN MAF < 0.1%) confer significantly greater absolute effect on standardized Lp(a) levels than variants observed in both ethnicities.

Third, WGS enables the detection of relevant genomic variants for Lp(a) which cannot be detected via WES or genotyping arrays. Furthermore, knowledge of such variants, given differential

effects on circulating Lp(a) and differential effects on incident cardiovascular events, provides additional information regarding cardiovascular disease risk beyond circulating Lp(a).

It should be noted that several limitations to this work exist. First, we estimate total KIV2-CN, but individuals may have different KIV2-CN alleles on each chromosome[42]. Our CNV analysis of next-generation sequencing data relies on aggregate depth of coverage for genotyping, precluding our ability to determine allelic KIV2-CN. However, despite this, sensitivity analyses suggest that the sum of KIV2-CN alleles may similarly associate with Lp(a) across varied KIV2-CN allele combinations. Additionally, the strongest SNP in our KIV2-CN imputation model is rs10455872, whose association with KIV2-CN has been well-described previously[17], and our KIV2-CN estimate is robustly associated with Lp(a) phenotypes as expected. Second, we only assess one non-European cohort; however, it has been observed that there are distinct Lp(a) distributions in other ethnicities which may uncover additional loci and sources of genetic heterogeneity. Furthermore, given the strong influence of ancestry on Lp(a), adjustment of *LPA* locus ancestry may improve power for genetic association. Indeed, prior analyses of African Americans suggest that genome-wide estimations of ancestry are correlated with *LPA* locus ancestry estimations[43]. Third, while in silico prediction tools for non-coding regions identify putative regulatory sequence, they are limited in their ability to (1) determine disruptive mutations, and (2) link regulatory regions to genes.

In summary, we characterize the shared and unique genetic determinants of Lp(a) using whole genome sequences in African Americans and Europeans. Additional knowledge of the complement of these determinants better informs cardiovascular disease risk prediction than biomarker alone.

## Methods

**Study participants**. Please refer to Supplementary Note 1 for study participant details. All study participants provided written and informed consent in accordance with respective institutional review boards for each of the participating study cohorts.

**WGS and variant calling**. Sequencing was performed at one of two sequencing centers, with all members within a cohort sequenced at the same center. The JHS WGS individuals were sequenced at University of Washington Northwest Genomics Center (Seattle, WA) as part of the as a part of the Phase 1 NIH/NHLBI Trans-Omics for Precision Medicine (TOPMed) program. The Finnish and Estonian WGS individuals were sequenced at the Broad Institute of Harvard and MIT (Cambridge, MA). Target coverage was >30× for JHS (mean attained 37.1), >20× for EST (mean attained 30.4), and >20× for FIN (mean attained 29.8).

TOPMED phase 1 BAM files were harmonized by the TOPMed Informatics Research Center (Center for Statistical Genetics, University of Michigan, Hyun Min Kang, Tom Blackwell and Goncalo Abecasis). In brief, sequence data were received from each sequencing center in the form of bam files mapped to the 1000 Genomes hs37d5 build 37 decoy reference sequence. Processing was coordinated and managed by the 'GotCloud' processing pipeline[44]. Samples with DNA contamination >3% (estimated using verifyBamId software[45]) and <95% of the genome covered at least 10× were filtered out. The JHS WGS used for analysis are from the "freeze 3a" genotype callsets of the variant calling pipeline performed using the software tools in the following repository: https://github.com/statgen/topmed_freeze3_calling, with variant detection performed by vt discover2 software tool[46].

WGS for FINRISK and the Estonian Biobank were performed using the Illumina HiSeqX platform at the Broad Institute of Harvard and MIT (Cambridge, MA). Libraries were normalized to 1.7 nM, constructed, and sequenced on the Illumina HiSeqX with the use of 151-bp paired-end reads for WGS and output was processed by Picard to generate aligned BAM files (to hg19)[47,48]. Variants were discovered using the Geome Analysis Tookit (GATK) v3 HaplotypeCaller according to Best Practices[49]. Finland and Estonia WGS samples were jointly called.

**Whole-genome sequence sample quality control**. The following three approaches were used by the TOPMed Genetic Analysis Center to identify and resolve sample identity issues in JHS: (1) concordance between annotated sex and biological sex inferred from the WGS data, (2) concordance between prior SNP array genotypes and WGS-derived genotypes, and (3) comparisons of observed and expected relatedness from pedigrees.

Additional measures for quality control of JHS, Finland, and Estonia were performed using the Hail software package (https://github.com/hail-is/hail)[50]. Samples were filtered by contamination (>3.0% for JHS, >5.0% for Finland and Estonia), chimeras >5%, GC dropout >4, raw coverage (<30× for JHS, <19× for Finland and Estonia), and indeterminate genotypic sex or genotypic/phenotypic sex mismatch (Supplementary Table 1).

**WGS genotype and variant quality control**. The variant filtering in JHS was performed by (1) first calculating Mendelian consistency scores using known familial relatedness and duplicates, and (2) training SVM classifier between the known variant sites (positive labels) and the Mendelian inconsistent variants (negative labels). Two additional hard filters were applied: (1) Excess heterozygosity filter (EXHET), if the Hardy–Weinberg disequilbrium $P$-value was less than $1 \times 10^{-6}$ in the direction of excess heterozygosity; (2) Mendelian discordance filter (DISC), with three or more Mendelian inconsistencies or duplicate discordances observed from the samples. Genotypes with a depth <10 were excluded, prior to filtering variants with >5% missingness.

Variants for Finland and Estonia were initially filtered by GATK Variant Quality Score Recalibration. Additionally, genotypes with GQ <20, DP <10 or >200, and poor allele balance (homozygous with <0.90 supportive reads or heterozygous with <0.20 supportive reads) were removed. Variants within low complexity regions were removed across all samples[51]. Variants with >20% missing calls, quality by depth <2 (SNPs) or <3 (indels), InbreedingCoeff <−0.3, and pHWE <1 × 10^{−9} were filtered out.

**Finnish imputation and quality control**. The imputation of the FINRISK samples[52] was done utilizing population specific reference panel of 2690 high-coverage whole-genome and 5093 high-coverage whole-exome sequences with IMPUTE2[53] that allows the usage of two panels at the same time. Before phasing and imputation, the data was QCed using following criteria: exclude samples with obscure sex, missingness (>5%), excess heterozygosity (+-4sd), non-European ancestry and SNPs with low call-rate (>2% missing), low HWE $P$-value (<1e-6), minor allele count (MAC) <3 (in case Zcalled[54]) or MAC <10 (if only called using Illumina GenCall). The haplotypic phase was determined using SHAPEIT2.0[55] prior to imputation. The FINRISK samples have been genotyped using multiple different genotyping chips, for which the QC, phasing and imputation was done in multiple chip-wise batches.

**Lp(a) and Lp(a)-C phenotypes**. Serum Lp(a)-C was measured in both EST and JHS via density gradient ultracentrifugation (Vertical Auto Profile [VAP], Atherotech).

Lp(a) was measured in JHS using a Diasorin nephelometric assay on a Roche Cobas FARA analyzer (Roche Diagnostics Corporation, Indianapolis, IN, USA), which measures Lp(a) mass by immunoprecipitin analysis using the SPQTM Antibody Reagent System of DiaSorin (DiaSorin Inc., Stillwater, MN 55082-0285). Turbidity produced by the antigen–antibody complexes was measured using the Roche Modular P Chemistry Analyzer. In FIN, Lp(a) was measured from serum stored at –70 °C using a commercially available latex immunoassay on an Architect c8000 system (Quantia Lp(a), Abbott Diagnostics).

Lp(a)-C and Lp(a) were inverse-rank normalized separately by cohort for analysis.

**Conventional lipid phenotypes**. Conventional lipoprotein cholesterols (HDL, LDL, TG, Total Cholesterol) and proteins (ApoB, ApoAI) were measured in EST and JHS by the VAP assay (where LDL refers to directly measured LDL, and not calculated). In FIN, these lipoproteins were measured via NMR as described in the MR methods below. In FIN, LDL cholesterol was either calculated by the Friedwald equation when triglycerides were <400 mg/dl or directly measured. Given the average effect of statins, when statins were present, total cholesterol was adjusted by dividing by 0.8 and LDL cholesterol by dividing by 0.7, as previously done[56]. All lipids were inverse-rank normalized separately by cohort in analysis.

**KIV2-CN estimation from WGS data**. Genome STRiP[21] version 2.00.1710 was used to estimate KIV2-CN in the *LPA* gene. Specifically, we ran Genome STRiP read-depth genotyping on the hg19 interval 6:161032614–161067851 using the following custom settings to capture an aggregate read-depth signal over every base position: -P depth.minimumMappingQuality:0, without specifying any of the usual genome masks.

After genotyping, we estimated the number of KIV2 protein domains from the raw copy number estimate by dividing the VCF genotype field CNF by the info field GSM1 and then estimating the KIV2 copy number by

$$KIV2 - CN = (CNF/GSM1) * 6.354 - 0.708$$

where 6.354 is derived from the number of full copies of the repeating unit represented on the hg19 reference genome and −0.708 is to adjust to the KIV2 units as visualized in Supplementary Fig. 6a, removing the outermost flanking exons that are part of the KIV1 and KIV3 (which are picked up in Genome STRiP due to their homology with the exons within the KIV2 domain).

**Evaluation of KIV2-CN precision**. To evaluate the precision of our measurements of KIV2 copy number, we utilized 123 pairs of siblings from JHS that were confidently IBD2 (identical-by-descent on both haplotypes) at the *LPA* locus. To identify these sibling pairs, we interrogated the hg19 interval 6:160,450,001–161,590,000 (0.5 Mb upstream and downstream of the *LPA* gene) and computed the concordance of SNP genotypes in this interval between all sequenced sibling pairs. We classified all sibling pairs with less than 1% genotype discordance as confidently IBD2 at the *LPA* locus and compared IBD2 sibling KIV2-CNs.

**KIV2-CN Imputation**. We split the FIN WGS into one training dataset comprised of two thirds of the samples (1477 samples) and one validation dataset (738 samples), and used the least absolute shrinkage and selection operator (LASSO), a machine-learning regression analysis method, using variants (using --indep-pairwise 50 5 0.25 in PLINK[57]) in a 4MB window around *LPA* imputed with high-quality (imputation quality >0.8) and MAF >0.001 in the FIN dataset. After applying 10-fold cross validation to find the optimal lambda (degree of shrinkage), the LASSO model selected 61 variants which minimized the mean squared error (Supplementary Fig. 8a). These 61 variants were also used in a random forest model to quantify the relative importance of each variant in the model (Supplementary Fig. 8b, Fig. 2b).

**Principle component analysis (PCA)**. To visualize PCs across all three cohorts against each other, a panel of approximately 16,000 ancestry informative markers[58] (AIMs) identified across six continental populations[59] was chosen to derive principal components (PCs) of ancestry for all samples that passed quality control. Principal component analysis was performed using EIGENSTRAT, using suggested quality control criteria[60] (Supplementary Fig. 3). Separately, within-cohort PCA was performed for use as covariates in analysis.

**Variant annotation**. Variants were annotated with Hail[50] using annotations from Ensembl's Variant Effect Predictor (VEP), ascribing the most severe, canonical consequence and gene to each variant[61]. For non-coding regions (in adult liver cells (E066), we used the Reg2Map HoneyBadger2-intersect[34] at strong ($P < 1 \times 10^{-10}$) DNase I hypersensitive regions (https://personal.broadinstitute.org/meuleman/reg2map/HoneyBadger2-intersect_release/).

Variants overlapping putative enhancers and promoters from the 25-state chromatin model[34] at this link were annotated and used in the single variant results annotations (Supplementary Data 2, 3), as well as grouping rare variants in the "sliding window" and "by distance" non-coding rare variant studies. Variants within 1MB of a known locus from the main lipids (LDL, HDL, TG, TC), as listed in Supplementary Data 15, were annotated as "KnownLocus_rsID" and "KnownLocus_Gene" within the single variant summary results files in Supplementary Data 2, 3.

**Single variant association**. Single variant analysis for EST and JHS WGS was performed using Hail's linear mixed-model regression[50] for associating each variant site with inverse normal transformed Lp(a) and Lp(a)-C within each cohort. All analyses were adjusted for KIV2-CN, age, sex, and an empirically derived kinship matrix to account for both familial and more distant relatedness[62]. To create the kinship matrix, regions of high-complexity known to have high LD were removed (as in the EPACTS make-kin --remove-complex flag); these regions included: 5:44000000–52000000, 6:24000000–36000000, 8:8000000–12000000, 11:42000000–58000000, and 17:40000000–43000000. Ten-fold random down-sampling of variants was performed to further reduce variant counts for fast processing-time.

For the FIN imputation dataset, single variant analysis was performed using SNPTEST (v2.5.2), using KIV2-CN, age, sex, fasting > 10 h, and adding PC1-10 as covariates to account for population structure due to absence of kinship matrix.

To ensure robust results, we only performed single variant analysis for variants with a MAF >0.001 within either cohort. Summary statistics for JHS and FIN for Lp(a) and JHS and EST for Lp(a)-C, for the corresponding inverse-rank normalized phenotypes, were meta-analyzed across cohorts using METAL[63], while also calculating heterogeneity statistics. Statistical significance alpha of $5 \times 10^{-8}$ was used for these analyses.

Additionally, for the *LPA* locus, iterative conditional association analysis was performed by cohort. Iterative conditioning was performed until $P > 5 \times 10^{-8}$ was attained.

**Heritability analyses**. Heritability analyses in EST WGS (for Lp(a)-C) and JHS WGS (for both Lp(a) and Lp(a)-C) were performed using Hail's linear mixed-model regression heritability estimate[50], described here https://hail.is/hail/hail.VariantDataset.html?highlight=lmm#hail.VariantDataset.lmmreg. Several filters were applied before variants were used in the kinship matrix. First, genome-wide variants underwent two-fold LD pruning as previously described via BOLT-REML[64], using variants with MAF > 0.001 and missingness < 1% with maximum LD $r^2 = 0.9$ (PLINK[57] commands used: --maf 0.001 --geno 0.01 --indep-pairwise 50 5 0.9). Regions of high-complexity were removed as previously described for single variant analysis. Ten-fold random down-sampling of variants was performed to further reduce variant counts for feasible analysis processing-time. For the heritability

estimates provided, 6,370,696 variants were used towards the kinship matrix in EST Lp(a)-C analysis, 1,897,407 variants in JHS Lp(a)-C analysis, and 1,894,291 variants in the JHS Lp(a) analysis. Baseline covariates used in the model, performed separately by cohort, included age, sex, fasting >10 h, and for EST, sequencing batch. A separate heritability estimate was also derived additionally conditioning on KIV2-CN.

For the FIN imputation dataset, variants were similarly limited, filtering for variants with MAF > 0.001, imputation quality > 0.8, and applying two-fold LD-pruning and removal of complex regions as described above (though the ten-fold down-sampling was not applied to keep the variant count on the same order of magnitude as in the WGS heritability analyses). A total of 3,088,864 variants were used towards heritability analysis, which was performed using BOLT-REML. Covariates used in the analysis included age, sex, fasting >10 h, and PC1-10. A separate heritability estimate was also derived additionally conditioning on KIV2-CN. For Lp(a), heritability analysis additionally conditioning on both KIV2-CN and the KIV2-CN-independent GRS using in MR was performed. BOLT-REML was also applied towards the Lp(a) heritability analysis in JHS, arriving at the same heritability estimates as Hail (data not shown).

**KIV2-CN modifier analysis**. Variant-by-KIV2-CN interaction analysis in the WGS was performed at a ~4MB window (6:158532140–162664257) around *LPA* to identify variants, which modify the relationship between directly genotyped KIV2-CN and Lp(a)-C (for EST and JHS) and Lp(a) (for JHS only). Variants with minor allele count >20 (by cohort) were included in analyses. The following interaction model was performed:

$$Lp(a)-C \sim KIV2 - CN + Variant + KIV2 - CN \times Variant + covariates$$

Where the interaction effect and *P*-value corresponds to the term: "KIV2-CN × Variant". Cohort-specific analyses were performed and for Lp(a)-C, EST and JHS interaction results were meta-analyzed using METAL[63]. Using the full interaction results, three top modifier variants were identified (rs13192132, rs1810126, and rs1740445) that were genome-wide significant upon meta-analysis ($P < 5 \times 10^{-8}$), in linkage equilibrium ($r^2 < 0.1$) across both ethnic backgrounds, and had replicating interaction effect directions in both ethnicities. To determine the cohort-specific Bonferroni significance threshold, LD clumping was performed on the full interaction results separately by cohort using the following PLINK[57] flags: --clump-kb 500 --clump-p1 1 --clump-p2 1 --clump-r2 0.25. In JHS, 1373 LD-pruned variants were identified, leading to a significance threshold of $P = 3.64 \times 10^{-5}$. In EST, 566 LD-pruned variants were identified, leading to a significance threshold of $P = 8.83 \times 10^{-5}$. Clumped variants with interaction p values surpassing the Bonferroni threshold are provided by cohort and phenotype in Supplementary Data 5. Overlap with methylation and acetylation marks was visualized using data from Roadmap for E066 adult liver cells at http://egg2.wustl.edu/roadmap/data/byFileType/alignments/consolidated/. Liver ATAC-seq data was downloaded from the ENCODE data portal (accession ENCFF893CSN). FASTQ files were adapter-trimmed and aligned to hg19 with bowtie2, and duplicates reads and reads with MAPQ <30 were removed.

Previous publications of variant-by-variant interactions have recommended performing sensitivity analyses to ensure significant interactions identified are not (1) due to the variants being in LD on the same haplotype and (2) mitigated by a separate third variant which explains the entire association[32,33]. In particular, the most recent study by Fish et al.[28] recommended that variant-by-variant interactions be performed using un-correlated variants (LD $r^2 < 0.6$). Thus, we checked the correlation of each of the three top identified variants with KIV2-CN by cohort (Supplementary Table 8), finding that these variants are indeed not correlated with KIV2-CN (Pearson correlation $r^2 < 0.1$). Furthermore, variants not associated ($P > 0.05$) with the phenotype are suggested to be removed, under the hypothesis that they may represent weak marginal effects from a true underlying interaction. Indeed, our three top Lp(a)-C interaction variants are all individually associated with Lp(a)-C (Supplementary Table 9). Lastly, conditional analysis has been suggested to ensure that the interaction model is not mitigated by a separate third variant that explains the interaction. Thus, we performed conditional analysis on the top three interaction models, conditioning on the previously identified variants from single variant analysis (reported in Supplementary Table 9) found to be conditionally independently associated with Lp(a)-C in each cohort. As seen in Supplementary Table 9, conditional analysis does not fully mitigate any of the identified interaction associations. Details on additional supplementary analysis performed imputing KIV2-CN using variants from the Illumina OmniQuad genotyping array is provided in Supplementary Note 3.

**Rare variant coding and non-coding association analyses (RVAS)**. Please refer to the Supplementary Note 4 for details on the coding and non-coding grouping schemes used. We tested the association of the aggregate of the aforementioned groupings with each lipid trait using the mixed-model Sequence Kernal Association Test (SKAT) implementation in EPACTS to account for bidirectional effects.[62] Analyses were adjusted for age, sex, fasting >10 h, sequencing batch (just used in Estonia), and empiric kinship. Groups with at least two rare variants and combined MAF >0.001 across all aggregated variants in a given cohort were included in meta-analysis. *P* values were meta-analyzed using Fisher's method. Statistical significance

for each RVAS test was based on the number of groups tested and is provided in the headers of Supplementary Data 6–13.

**Mendelian randomization**. We developed three genetic instruments per cohort. The first instrument used was a genetic risk score, "GRS," comprised of variants in a ~4MB window around *LPA* (6:158532140–162664257) with sub-threshold significance ($P$-value $< 1 \times 10^{-4}$), using variant effect sizes from the KIV2-CN conditioned single variant analysis and performing LD clumping in plink using the following parameters: --clump-kb 500 --clump-p1 0.0001 --clump-p2 1 --clump-r2 0.25. This resulted in 399 variants for Lp(a) GRS in FIN, 235 variants for Lp(a) GRS in JHS, 39 variants for Lp(a)-C GRS in JHS, and 49 variants for Lp(a)-C GRS in EST (Supplementary Data 14). The second instrument used was a "KIV2-CN" score using the directly genotyped or imputed KIV2-CN. The third instrument used was a combined "GRS + KIV2-CN" score combining scores from (1) and (2). Each of the three scores were inverse rank normalized and adjusted such that 1 unit increase in the score is equal to 1 SD increase in Lp(a) (or Lp(a)-C, depending on how the instrument was adjusted). The multiplicative factors used to adjust each score are provided in Supplementary Table 10.

Please refer to Supplementary Note 2 for details on additional MESA, FHS, and OOA participants used in subclinical atherosclerosis instrumental variable analyses. The Lp(a) GRS for Europeans in MESA and FHS was based off of the FIN Lp(a) GRS, the Lp(a) GRS for African Americans in MESA and JHS was based off of the JHS Lp(a) GRS, the Lp(a)-C GRS for Europeans in MESA, FHS, and OOA was based off of the EST Lp(a)-C GRS, and the Lp(a)-C GRS for African Americans in MESA and JHS was based off of the JHS Lp(a)-C GRS.

Please refer to Supplementary Note 5 for details on incident events and subclinical measures used. For incident clinical events, a cox proportional hazards test was performed, finding the association between each incident event and each of the genetic instruments, as well as observational Lp(a). For the quantitative subclinical measures, linear regression was performed, finding the association between each inverse-rank normalized phenotype and each of the genetic instruments, as well as inverse-rank normalized Lp(a) and Lp(a)-C (where available). Covariates used in all analyses included the first five principal components of genetic ancestry, age, sex, if the individual was fasting >10 h. Statistical significance was defined for the 10 FIN incident clinical events and two subclinical atherosclerosis traits using a Bonferroni significance threshold was based on the number of outcome phenotypes analyzed ($P = 0.005$ and 0.025, respectively).

**Data availability**. Individual-level genotype and phenotype information for TOPMed studies are available in dbGAP (JHS: phs000964, FHS: phs000974, MESA: phs001416, OOA: phs000956). Summary-level list of genotypes and genotype counts are available on the BRAVO server (https://bravo.sph.umich.edu/). The Finnish WGS and array genotype data can be accessed through THL Biobank (https://thl.fi/fi/web/thl-biobank). The WGS data at Estonian Genome Center, University of Tartu can be accessed via Estonian Biobank (www.biobank.ee).

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

## Acknowledgements

Please refer to Supplementary Note 6 for Acknowledgements.

## Author contributions

S.M.Z., P.N., A.G., I.S., and S.K. designed the study; S.M.Z., S.R., R.E.H., G.M.P., A.M., C. Y., and K.A.R. performed the analyses; S.M.Z., S.R., R.E.H., P.N., S.K., S.R., R.E.H., M.A., M.J.D., I.S., A.G. performed interpretation of data; S.M.Z., P.N., A.G., and S.K. drafted the manuscript; S.M.Z., P.N., A.G., S.K., I.S., R.E.H., M.A., V.S., J.G.W., S.R., and M.J.D. revised the manuscript; J.B., T.P., and C.S. provided Hail software support; J.Ernst, J. Engreitz, and M.C. provided other technical support; and A.C., A.M., V.S., M.K., M.J.D, J. G.W., B.M.N., S.M., I.S., T.E., S.R., S.K., M.T., M.B., V.S.R., L.A.C., W.C.J., J.I.R., S.S.R., W.P., B.D.M., M.F., and P.N. provided administrative or material support.

## Additional information

**Competing interests:** The authors declare no competing interests.

Seyedeh M. Zekavat [1,2,3], Sanni Ruotsalainen[4], Robert E. Handsaker [1,5,6], Maris Alver[8,9], Jonathan Bloom [1,7], Timothy Poterba[1,7], Cotton Seed[1,7], Jason Ernst [10], Mark Chaffin[1], Jesse Engreitz [1], Gina M. Peloso[11], Ani Manichaikul[12], Chaojie Yang[12], Kathleen A. Ryan[13], Mao Fu[13], W. Craig Johnson[14], Michael Tsai[15], Matthew Budoff[16], Ramachandran S. Vasan[17,18], L. Adrienne Cupples[11,17], Jerome I. Rotter[19], Stephen S. Rich[12], Wendy Post[20], Braxton D. Mitchell[21], Adolfo Correa [22], Andres Metspalu[9], James G. Wilson[22], Veikko Salomaa[23], Manolis Kellis[1,24], Mark J. Daly[1,5,7], Benjamin M. Neale [1,5,7], Steven McCarroll[1,5,6], Ida Surakka[4], Tonu Esko [1,9], Andrea Ganna[1,5,7], Samuli Ripatti[1,4,25], Sekar Kathiresan [1,26,27,28] & Pradeep Natarajan [1,26,27,28], NHLBI TOPMed Lipids Working Group

[1]Program in Medical and Population Genetics, Broad Institute of MIT and Harvard, Cambridge, MA 02142, USA. [2]Yale School of Medicine, New Haven, CT 06510, USA. [3]Department of Computational Biology & Bioinformatics, Yale University, New Haven, CT 06510, USA. [4]Institute for Molecular Medicine, University of Helsinki, Helsinki, Finland. [5]Stanley Center for Psychiatric Research, Broad Institute of MIT and Harvard, Cambridge, MA 02142, USA. [6]Department of Genetics, Harvard Medical School, Boston, MA 02115, USA. [7]Analytic and Translational Genetics Unit, Boston, MA 02142, USA. [8]Department of Biotechnology, Institute of Molecular and Cell Biology, University of Tartu, Tartu, Estonia. [9]Estonian

Genome Center, Tallinn, Estonia. [10]Department of Biological Chemistry, University of California, Los Angeles, Los Angeles, CA 90095, USA. [11]Department of Biostatistics, Boston University School of Public Health, Boston, MA 02118, USA. [12]Center for Public Health Genomics, University of Virginia, Charlottesville, VA 22904, USA. [13]Program in Personalized and Genomic Medicine, Division of Endocrinology, Diabetes & Nutrition, Department of Medicine, University of Maryland School of Medicine, Baltimore, MD 21201, USA. [14]Department of Biostatistics, School of Public Health and Community Medicine, University of Washington, Seattle, WA 98195, USA. [15]Department of Laboratory Medicine and Pathology, University of Minnesota, Minneapolis, MN 55455, USA. [16]Division of Cardiology, Harbor-UCLA Medical Center, Los Angeles Biomedical Research Institute, Los Angeles, CA 90509, USA. [17]NHLBI Framingham Heart Study, Framingham, MA 20892, USA. [18]Sections of Preventive medicine and Epidemiology, and cardiovascular medicine, Departments of Medicine and Epidemiology, Boston university Schools of Medicine and Public health, Boston, MA 02118, USA. [19]Departments of Pediatrics and Medicine, The Institute for Translational Genomics and Population Sciences, Los Angeles Biomedical Research Institute, Harbor-UCLA Medical Center, Torrance, CA 90509, USA. [20]Division of Cardiology, Department of Medicine, Johns Hopkins University School of Medicine, Baltimore, MD 21205, USA. [21]Department of Medicine, University of Maryland School of Medicine, Baltimore, MD 21201, USA. [22]Department of Medicine, University of Mississippi Medical Center, Jackson, MS 39216, USA. [23]National Institute for Health and Welfare, Helsinki, Finland. [24]Computer Science and Artificial Intelligence Lab, Massachusetts Institute of Technology, 32 Vassar St, Cambridge, MA 02139, USA. [25]Department of Public Health, Faculty of Medicine, University of Helsinki, Helsinki, Finland. [26]Department of Medicine, Harvard Medical School, Boston, MA 02115, USA. [27]Center for Genomic Medicine, Massachusetts General Hospital, Boston, MA 02114, USA. [28]Cardiovascular Research Center, Massachusetts General Hospital, Boston, MA 02114, USA. These authors jointly supervised this work: Tonu Esko, Andrea Ganna, Samuli Ripatti, Sekar Kathiresan, Pradeep Natarajan.

## NHLBI TOPMed Lipids Working Group

Namiko Abe[29], Goncalo Abecasis[30], Christine Albert[31], Nicholette (Nichole) Palmer Allred[32], Laura Almasy[33,34], Alvaro Alonso[35], Seth Ament[36], Peter Anderson[37], Pramod Anugu[38], Deborah Applebaum-Bowden[39], Dan Arking[40], Donna K Arnett[41], Allison Ashley-Koch[42], Stella Aslibekyan[43], Tim Assimes[44], Paul Auer[45], Dimitrios Avramopoulos[40], John Barnard[46], Kathleen Barnes[47], R. Graham Barr[48], Emily Barron-Casella[40], Terri Beaty[40], Diane Becker[40], Lewis Becker[40], Rebecca Beer[39], Ferdouse Begum[40], Amber Beitelshees[36], Emelia Benjamin[49,31], Marcos Bezerra[50], Larry Bielak[30], Joshua Bis[37], Thomas Blackwell[30], John Blangero[51], Eric Boerwinkle[52], Ingrid Borecki[37], Russell Bowler[53], Jennifer Brody[37], Ulrich Broeckel[54], Jai Broome[37], Karen Bunting[29], Esteban Burchard[55], Jonathan Cardwell[47], Cara Carty[56], Richard Casaburi[57], James Casella[40], Christy Chang[36], Daniel Chasman[58], Sameer Chavan[47], Bo-Juen Chen[29], Wei-Min Chen[59], Yii-Der Ida Chen[60], Michael Cho[58], Seung Hoan Choi[61], Lee-Ming Chuang[62], Mina Chung[46], Elaine Cornell[63], Carolyn Crandall[57], James Crapo[53], Joanne Curran[51], Jeffrey Curtis[30], Brian Custer[64], Coleen Damcott[36], Dawood Darbar[65], Sayantan Das[30], Sean David[44], Colleen Davis[37], Michelle Daya[47], Mariza de Andrade[66], Michael DeBaun[67], Ranjan Deka[68], Dawn DeMeo[58], Scott Devine[36], Ron Do[69], Qing Duan[70], Ravi Duggirala[71], Peter Durda[63], Susan Dutcher[72], Charles Eaton[73], Lynette Ekunwe[38], Patrick Ellinor[31], Leslie Emery[37], Charles Farber[59], Leanna Farnam[58], Tasha Fingerlin[53], Matthew Flickinger[30], Myriam Fornage[52], Nora Franceschini[70], Stephanie M. Fullerton[37], Lucinda Fulton[72], Stacey Gabriel[61], Weiniu Gan[39], Yan Gao[38], Margery Gass[74], Bruce Gelb[69], Xiaoqi (Priscilla) Geng[30], Soren Germer[29], Chris Gignoux[44], Mark Gladwin[75], David Glahn[76], Stephanie Gogarten[37], Da-Wei Gong[36], Harald Goring[77], C. Charles Gu[72], Yue Guan[36], Xiuqing Guo[60], Jeff Haessler[74,56], Michael Hall[38], Daniel Harris[36], Nicola Hawley[76], Jiang He[78], Ben Heavner[37], Susan Heckbert[37], Ryan Hernandez[55], David Herrington[32], Craig Hersh[58], Bertha Hidalgo[43], James Hixson[52], John Hokanson[47], Elliott Hong[36], Karin Hoth[79], Chao (Agnes) Hsiung[80], Haley Huston[81], Chii Min Hwu[82], Marguerite Ryan Irvin[43], Rebecca Jackson[83], Deepti Jain[37], Cashell Jaquish[39], Min A Jhun[30], Jill Johnsen[81,37], Andrew Johnson[84], Rich Johnston[35], Kimberly Jones[40], Hyun Min Kang[30], Robert Kaplan[85], Sharon Kardia[30], Laura Kaufman[58], Shannon Kelly[64], Eimear Kenny[69], Michael Kessler[36], Alyna Khan[37], Greg Kinney[47], Barbara Konkle[81], Charles Kooperberg[74], Holly Kramer[86], Stephanie Krauter[37], Christoph Lange[87], Ethan Lange[47], Leslie Lange[47], Cathy Laurie[37], Cecelia Laurie[37], Meryl LeBoff[58], Seunggeun Shawn Lee[30], Wen-Jane Lee[82], Jonathon LeFaive[30], David Levine[37], Dan Levy[84], Joshua Lewis[36], Yun Li[70], Honghuang Lin[49], Keng Han Lin[30], Simin Liu[73,56], Yongmei Liu[32], Ruth Loos[69], Steven Lubitz[31], Kathryn Lunetta[49], James Luo[84], Michael Mahaney[51], Barry Make[40], JoAnn Manson[58], Lauren Margolin[61], Lisa Martin[88], Susan Mathai[47], Rasika Mathias[40], Patrick McArdle[36], Merry-Lynn McDonald[43], Sean McFarland[89], Stephen McGarvey[73],

Hao Mei[38], Deborah A Meyers[90], Julie Mikulla[39], Nancy Min[38], Mollie Minear[39], Ryan L Minster[75], May E. Montasser[36], Solomon Musani[38], Stanford Mwasongwe[38], Josyf C Mychaleckyj[59], Girish Nadkarni[69], Rakhi Naik[40], Sergei Nekhai[91], Deborah Nickerson[37], Kari North[70], Jeff O'Connell[36], Tim O'Connor[36], Heather Ochs-Balcom[92], James Pankow[93], George Papanicolaou[39], Margaret Parker[58], Afshin Parsa[36], Sara Penchev[53], Juan Manuel Peralta[71], Marco Perez[44], James Perry[36], Ulrike Peters[74,37], Patricia Peyser[30], Larry Phillips[35], Sam Phillips[37], Toni Pollin[36], Julia Powers Becker[47], Meher Preethi Boorgula[47], Michael Preuss[69], Dmitry Prokopenko[89], Bruce Psaty[37], Pankaj Qasba[39], Dandi Qiao[58], Zhaohui Qin[35], Nicholas Rafaels[47], Laura Raffield[70], D.C. Rao[72], Laura Rasmussen-Torvik[94], Aakrosh Ratan[59], Susan Redline[58], Robert Reed[36], Elizabeth Regan[53], Alex Reiner[74,37], Ken Rice[37], Dan Roden[67], Carolina Roselli[61], Ingo Ruczinski[40], Pamela Russell[47], Sarah Ruuska[81], Phuwanat Sakornsakolpat[58], Shabnam Salimi[36], Steven Salzberg[40], Kevin Sandow[60], Vijay Sankaran[89], Christopher Scheller[30], Ellen Schmidt[30], Karen Schwander[72], David Schwartz[47], Frank Sciurba[75], Christine Seidman[95], Vivien Sheehan[96], Amol Shetty[36], Aniket Shetty[47], Wayne Hui-Heng Sheu[82], M. Benjamin Shoemaker[67], Brian Silver[97], Edwin Silverman[58], Jennifer Smith[30], Josh Smith[37], Nicholas Smith[37], Tanja Smith[29], Sylvia Smoller[85], Beverly Snively[32], Tamar Sofer[58], Nona Sotoodehnia[37], Adrienne Stilp[37], Elizabeth Streeten[36], Yun Ju Sung[72], Jody Sylvia[58], Adam Szpiro[37], Carole Sztalryd[36], Daniel Taliun[98], Hua Tang[44], Margaret Taub[40], Kent Taylor[60], Simeon Taylor[36], Marilyn Telen[42], Timothy A. Thornton[37], Lesley Tinker[56], David Tirschwell[37], Hemant Tiwari[43], Russell Tracy[63], Dhananjay Vaidya[40], Peter VandeHaar[30], Scott Vrieze[93,99], Tarik Walker[47], Robert Wallace[79], Avram Walts[47], Emily Wan[58], Fei Fei Wang[37], Karol Watson[57], Daniel E. Weeks[75], Bruce Weir[37], Scott Weiss[58], Lu-Chen Weng[31], Cristen Willer[30], Kayleen Williams[37], L. Keoki Williams[100], Carla Wilson[58], Quenna Wong[37], Huichun Xu[36], Lisa Yanek[40], Ivana Yang[47], Rongze Yang[36], Norann Zaghloul[36], Yingze Zhang[75], Snow Xueyan Zhao[53], Wei Zhao[30], Xiuwen Zheng[37], Degui Zhi[52], Xiang Zhou[30], Michael Zody[29] & Sebastian Zoellner[30]

[29]New York Genome Center, New York, NY 10013, USA. [30]University of Michigan, Ann Arbor, MI 48109, USA. [31]Massachusetts General Hospital, Boston, MA 02114, USA. [32]Wake Forest Baptist Health, Winston-Salem, NC 27157, USA. [33]Children's Hospital of Philadelphia, University of Pennsylvania, Philadelphia, PA 19104, USA. [34]University of Pennsylvania, Philadelphia, PA 19104, USA. [35]Emory University, Atlanta, GA 30322, USA. [36]University of Maryland, Baltimore, MD 21201, USA. [37]University of Washington, Seattle, WA 98195, USA. [38]University of Mississippi, Jackson, MS 38677, USA. [39]National Institutes of Health, Bethesda, MD 20892, USA. [40]Johns Hopkins University, Baltimore, MD 21218, USA. [41]University of Kentucky, Lexington, KY 40506, USA. [42]Duke University, Durham, NC 27708, USA. [43]University of Alabama, Birmingham, AL 35487, USA. [44]Stanford University, Stanford, CA 94305, USA. [45]University of Wisconsin Milwaukee, Milwaukee, WI 53211, USA. [46]Cleveland Clinic, Cleveland, OH 44195, USA. [47]University of Colorado, Denver, CO, USA 80204. [48]Columbia University, New York, NY 10027, USA. [49]Boston University, Boston, MA 02215, USA. [50]Fundação de Hematologia e Hemoterapia de Pernambuco - Hemope, Recife 52011-000, Brazil. [51]University of Texas Rio Grande Valley School of Medicine, Brownsville, TX 78520, USA. [52]University of Texas Health, Houston, TX 77225, USA. [53]National Jewish Health, Denver, CO 80206, USA. [54]Medical College of Wisconsin, Milwaukee, WI 53226, USA. [55]University of California, San Francisco, San Francisco, CA 94143, USA. [56]Women's Health Initiative, Seattle, WA 98109, USA. [57]University of California, Los Angeles, Los Angeles, CA 90095, USA. [58]Brigham & Women's Hospital, Boston, MA 02115, USA. [59]University of Virginia, Charlottesville, VA 22903, USA. [60]Los Angeles Biomedical Research Institute, Los Angeles, CA 90502, USA. [61]The Broad Institute, Cambridge, MA 02142, USA. [62]National Taiwan University, Taipei 10617, Taiwan. [63]University of Vermont, Burlington, VT 05405, USA. [64]Blood Systems Research Institute UCSF, San Francisco, CA 94118, USA. [65]University of Illinois at Chicago, Chicago, IL 60607, USA. [66]Mayo Clinic, Rochester, MN 55905, USA. [67]Vanderbilt University, Nashville, TN 37235, USA. [68]University of Cincinnati, Cincinnati, OH 45220, USA. [69]Icahn School of Medicine at Mount Sinai, New York, NY 10029, USA. [70]University of North Carolina, Chapel Hill, NC 27599, USA. [71]University of Texas Rio Grande Valley School of Medicine, Edinburg, TX 78539, USA. [72]Washington University in St Louis, St Louis, MO 63130, USA. [73]Brown University, Providence, RI 02912, USA. [74]Fred Hutchinson Cancer Research Center, Seattle, WA 98109, USA. [75]University of Pittsburgh, Pittsburgh, PA 15260, USA. [76]Yale University, New Haven, CT 06520, USA. [77]University of Texas Rio Grande Valley School of Medicine, San Antonio, TX 78229, USA. [78]Tulane University, New Orleans, LA 70118, USA. [79]University of Iowa, Iowa City, IA 52242, USA. [80]National Health Research Institute Taiwan, Zhunan Township 350, Taiwan. [81]Blood Works Northwest, Seattle, WA 98105, USA. [82]Taichung Veterans General Hospital Taiwan, Taichung City 407, Taiwan. [83]Ohio State University Wexner Medical Center, Columbus, OH 43210, USA. [84]NIH National Heart, Lung, and Blood Institute, Bethesda, MD 98106, USA. [85]Albert Einstein College of Medicine, New York, NY 20892, USA. [86]Loyola University, Maywood, IL 10461, USA. [87]Harvard School of Public Health, Boston, MA 98104, USA. [88]George Washington University, Washington 60153, USA. [89]Harvard University, Cambridge, MA 02115, USA. [90]University of Arizona, Tucson, AZ 20052, USA. [91]Howard University, Washington 02138, USA. [92]University at Buffalo, Buffalo, NY 85721, USA. [93]University of Minnesota, Minneapolis, MN 20059, USA. [94]Northwestern University, Chicago, IL 14260, USA. [95]Harvard Medical School, Boston, MA 55455, USA. [96]Baylor College of Medicine, Houston, TX 60208, USA. [97]UMass Memorial Medical Center, Worcester, MA 98107, USA. [98]Baylor College of Medicine, Ann Arbor, MI 02115, USA. [99]University of Colorado at Boulder, Boulder, CO 77030, USA. [100]Henry Ford Health System, Detroit, MI 01655, USA

