## [Peer Review File · Nature Communications]

Reviewer #1 (Remarks to the Author):

This paper describes a genetic association study of Lp(a), a highly heritable and causal risk factor for cardiovascular disease. The authors use a single assay deep coverage whole genome sequencing approach in 8,392 individuals of European and African ancestry to directly genotype KIV2-CN from sequence data. This has not been done before. Most interestingly, they provide a new model for KIV2-CN imputation. A highlight of biomedical interest is the identification of an association at the SORT1 coronary heart disease risk locus with Lp(a) cholesterol, independent of LDL cholesterol. The paper is well written and clear. It provides an extensive set of supplemental data that allows to retrace arguments for even the most minor points down to the last detail. I have only very minor remarks and optional suggestions.

Abstract: The authors write that “LPA genotype is more strongly associated with incident cardiovascular diseases than directly measured Lp(a)”. I am not sure how relevant this statement is. The strength of an association is in part dependent on the measurement error of the variable in question. Could a more precise measurement method for Lp(a) change this conclusion?

Line 124/125: The authors observe a 10-fold difference between JHS and FIN Lp(a) concentrations. However, they are using different measurement platforms for Lp(a) in these studies. Can you comment on how much of this difference may potentially be due to technical differences? I assume it would be minor, but it would be good briefly mention it.

Line 154 & Fig 2c: The authors present a 61-variant model to impute KIV2-CN. This imputation represents a major advance of this work. However, the large number of SNPs makes implementation of the method in other cohorts also potentially difficult if not all SNPs are available. It would be useful if the authors could report the performance of a model that would use only, say, the 5 strongest SNPs, or a small number of SNPs that are directly available from common genotyping platforms (why not SNPs genotyped by 23andMe?). This would provide an easy way for less sophisticated studies to obtain a quick estimate of KIV2-CN.

Fig 4d: Is this panel needed? Consider reporting the numbers in the text.

Remark: Regarding the discussion of the SORT1 locus, the authors may be interested in the fact that Sun et al. report an association of SNP rs4970836 with a number of proteins, including Group XIIB secretory phospholipase A2-like protein (PLA2G12B). rs4970836 is in high LD with rs12740374 ($r^2=0.93$) – see Supplemental Table 1, Line 51 of <https://www.biorxiv.org/content/early/2017/05/05/134551>

Reviewer #2 (Remarks to the Author):

Suppl figure 3 shows as expected that the JHS African Americans are variably admixed. The inclusion of genome-wide derived PCs as covariates cannot guarantee that potential confounding is eliminated, as an individual's local (e.g. to LPA) ancestry may obviously differ from their "average ancestry across the entire genome". This is especially important here as $Lp(a)$ levels vary so much between the African and European ancestry groups as documented on lines 124 & 125; a "perfect storm for stratification". Ideally the authors would refit their association models allowing for local-to-LPA ancestry in JHS but at minimum I recommend that this "limitation of inference" is noted in the discussion.

Line 128 and Suppl Figure 5: A " $0.3 < \rho < 0.5$ " is typically referred to as a moderate correlation, the p-value is tiny more due to the huge sample size than the magnitude of ρ .

Line 144/145: ρ (Figure 7c) or ρ^2 (line 145); this I suggest is a "very strong" (magnitude of ρ) as well as a "robust" (to outliers) non-parametric correlation.

Lines 158 & 159: Fig 2c (not Fig 3b?) shows a strong correlation between genotyped and imputed KIV2-CN, but the regression slope is < 1 , add a $y=x$ reference line to highlight this please (as I don't think many in the field are aware of this feature of genotype imputation).

Line 242: Confirm ρ or ρ^2 ?

Line 305: "10 incident clinical events" means "10 different clinical phenotypes", not $n=10$ patients? Rephrase?

Lines 309 – 311: This introduces a key result, with the umbrella "cardiovascular disease" model selected from the 40 results listed in Suppl. Table 30, perhaps as it shows the maximal discrepancy in HR contrasting GRS vs. KIV2-CV? Fig. 7a does though show a similar, albeit slightly less extreme, difference in HR for the two predictors for the MI or the slightly broader CHD phenotype. These latter two phenotypes are arguably "cleaner" and less pathophysiologically heterogeneous than "cardiovascular disease", and have been widely used in GWAS with great success. So I suggest that results drawn from Fig. 7a are sufficient to satisfactorily make the point here.

Line 925: I think you mean Suppl Fig 8a (& 8b) (not Suppl Fig 6a). Would be helpful to show the MSE for >61 variants in Suppl. Fig 8b to fully appreciate the stopping decision.

Response to Reviewers

Reviewer #1:

This paper describes a genetic association study of Lp(a), a highly heritable and causal risk factor for cardiovascular disease. The authors use a single assay deep coverage whole genome sequencing approach in 8,392 individuals of European and African ancestry to directly genotype KIV2-CN from sequence data. This has not been done before. Most interestingly, they provide a new model for KIV2-CN imputation. A highlight of biomedical interest is the identification of an association at the SORT1 coronary heart disease risk locus with Lp(a) cholesterol, independent of LDL cholesterol. The paper is well written and clear. It provides an extensive set of supplemental data that allows to retrace arguments for even the most minor points down to the last detail. I have only very minor remarks and optional suggestions.

Author Response:

We thank Reviewer #1 for their thoughtful comments. In addition to addressing the reviews below, we have also added additional individuals of African and European ancestries (from the Multi-Ethnic Study of Atherosclerosis, Old Order Amish, and Framingham Heart Study) to the subclinical atherosclerosis Mendelian randomization analysis (to assess *LPA* variant classes with each CAC and AAC). These additional analyses are now described in the text and via Fig. 7b (see updated panel below), Supplementary Fig. 28, and Supplementary Table 31. Co-authors from each cohort have been added accordingly for their contributions.

Manuscript Amendment:

Please see new/revised Figure 7B, Supplementary Figure 28, and Supplementary Table 31. Figure 7B is pasted below.

Additionally, the text is updated to reflect the inclusion of additional cohorts to these analyses.

Abstract:

1. The authors write that “LPA genotype is more strongly associated with incident cardiovascular diseases than directly measured Lp(a)”. I am not sure how relevant this statement is. The strength of an association is in part dependent on the measurement error of the variable in question. Could a more precise measurement method for Lp(a) change this conclusion?

Author Response:

We agree that, in addition to sample size, strength of association for these measures is dependent on estimated effect and precision of that estimate. The precision can be influenced by precision of Lp(a) measures. However, intraindividual temporal variability of Lp(a) is modest (PMID: 29174389). We highlight here that the estimated effect conferred per standard deviation of each measure (Lp(a) biomarker vs LPA genotype score) is different; we have now clarified this key distinction in the next. The hazard ratios for association of Lp(a) with incident coronary heart disease is HR 1.16/Lp(a) SD ($P = 3.7 \times 10^{-3}$) and for LPA GRS with incident coronary heart disease is HR 1.36/genetic Lp(a) SD ($P = 7.6 \times 10^{-8}$).

Manuscript Amendment:

“LPA risk genotypes confer greater relative risk for incident atherosclerotic cardiovascular diseases compared to directly measured Lp(a)”

2. Line 124/125: The authors observe a 10-fold difference between JHS and FIN Lp(a) concentrations. However, they are using different measurement platforms for Lp(a) in these studies. Can you comment on how much of this difference may potentially be due to technical differences? I assume it would be minor, but it would be good briefly mention it.

Author Response:

Both JHS and FIN samples were measured using immunoassay-based methods for measuring Lp(a) particle mass. As they were performed on different commercial assays, there may be systematic differences but we don't anticipate a 10-fold difference to be explained by this technical difference. Similar systemic Lp(a) distribution differences have been observed between diverse ethnicities (PMID: 19060253, 20160194). Further, Northern Europeans have been previously observed to have the lowest Lp(a) concentrations across the European population. Reference 20 in the main text (PMID: 28449027) notes: “We found lower Lp(a) levels in Northern European cohorts (median 4.9 mg/dL) compared to central (median 7.9 mg/dL) and Southern European cohorts (10.9 mg/dL) (Jonckheere–Terpstra test $P < 0.001$).” The former Lp(a) median is similar to the mean observed in the present study's Northern Europeans (i.e., FIN) (5.0 mg/dL).

To minimize influences of population stratification in analyses, we normalized all phenotypes by Lp(a) assay as well as by cohort.

Manuscript Amendment:

“Finnish individuals have among the lowest Lp(a) concentrations across European populations.²⁰ This may explain why we observe a 10-fold difference between JHS and FIN Lp(a) concentrations versus the 2-3 fold differences previously observed between African and European populations¹⁶.”

“Lp(a) values were quantified using two immunoassay-based methods sensitive to the entire mass of the Lp(a) particle.”

3. Line 154 & Fig 2c: The authors present a 61-variant model to impute KIV2-CN. This imputation represents a major advance of this work. However, the large number of SNPs makes implementation of the method in other cohorts also potentially difficult if not all SNPs are available. It would be useful if the authors could report the performance of a model that would use only, say, the 5 strongest SNPs, or a small number of SNPs that are directly available from common genotyping platforms (why not SNPs genotyped by 23andMe?). This would provide an easy way for less sophisticated studies to obtain a quick estimate of KIV2-CN.

Author Response:

We agree with the reviewer that many cohorts will not have the capability to directly genotype KIV2-CN from next generation sequence data and that much larger numbers of samples have array-derived genotypes where KIV2-CN may be potentially imputed. With these issues in mind, we developed a 61-SNP KIV2-CN imputation model for cohorts with genotyping arrays and further dense SNP imputation using reference panels.

The reviewer highlights that many cohorts may further not have additional SNP imputation to leverage our 61-SNP KIV2-CN imputation approach. Ability to use the full 61 SNPs (or proxies) will provide the most optimal estimation with logarithmic decay using fewer SNPs, as depicted in Supplementary Fig. 8B (copied below for reference). For example, the mean-squared error in KIV2-CN prediction is nearly 2-fold greater with 5 variants compared to 61 variants.

To further explore the model’s performance using variants present in a conventional genotyping array platform, we ascertained the variants or suitable proxies (of the 61 SNPs) on the Illumina OmniQuad genotyping array. 6 of the 61 variants (including the top two variants) were either present or had proxies (with LD $r^2 > 0.8$) available in the OmniQuad array. Using

these variants, KIV2-CN may be imputed but with less precision (Pearson $r = 0.62$, explaining 38% of variation in KIV2-CN). While we recommend using all 61 variants (or their proxies) to maximize predictive performance, we have also included this additional analysis in the supplementary text.

Manuscript Amendment:

“Imputation of KIV2-CN using variants from the Illumina OmniQuad genotyping array

To further explore the model’s performance using variants present in a conventional genotyping array platform, we determined the overlap between the 61 variants and variants within the Illumina OmniQuad genotyping array using SNAP (<http://archive.broadinstitute.org/mpg/snap/ldsearch.php>). 6 of the 61 variants (including the top two important variants) were either present or had proxies (with LD $r^2 > 0.8$) available in the OmniQuad array. Using these variants, we re-computed LASSO coefficients (listed below) using the same methodology as previously described and found the predictive performance to be lower than the 61-variant model, with Pearson coefficient of 0.62 between estimated and genotyped KIV2-CN and explaining 38% of variation in KIV2-CN.

Variant (hg19 chr.pos.ref.alt)	rsID	OmniQuad Variant in LD	LASSO Coefficient
6.160910517.T.A	rs12214416	rs12214416 ($r^2 = 1$)	1.62
6.160919223.T.C	rs4129086	rs4129086 ($r^2 = 1$)	0.91
6.161010118.A.G	rs10455872	rs10455872 ($r^2 = 1$)	-9.20
6.161068320.C.G	rs4708876	rs7770628 ($r^2 = 0.87$)	-2.91
6.161068607.T.C	rs12526465	rs12526465 ($r^2 = 1$)	-8.17

6.161233297.C.T	rs117774213	rs9458173 ($r^2 = 1$)	10.41
(Intercept)	-	-	49.48

4. Fig 4d: Is this panel needed? Consider reporting the numbers in the text.

Author Response:

This panel is included to highlight a key point: the high heritability for this CHD causal biomarker across populations. Prior estimates of heritability have had marked variability; our estimates are closer to estimates for twin studies.

5. Remark: Regarding the discussion of the SORT1 locus, the authors may be interested in the fact that Sun et al. report an association of SNP rs4970836 with a number of proteins, including Group XIIB secretory phospholipase A2-like protein (PLA2G12B). rs4970836 is in high LD with rs12740374 ($r^2=0.93$) – see Supplemental Table 1, Line 51 of <https://www.biorxiv.org/content/early/2017/05/05/134551>

Author Response:

We appreciate the reviewer's remarks and agree that additional pleiotropy analyses for this locus and others relevant to the present study may uncover important biological insights in future studies.

Reviewer #2:

We thank Reviewer #2 for their thoughtful comments and ask that they also refer to the comments under the main remarks from Reviewer #1.

1. Suppl figure 3 shows as expected that the JHS African Americans are variably admixed. The inclusion of genome-wide derived PCs as covariates cannot guarantee that potential confounding is eliminated, as an individual's local (e.g. to *LPA*) ancestry may obviously differ from their “average ancestry across the entire genome”. This is especially important here as $Lp(a)$ levels vary so much between the African and European ancestry groups as documented on lines 124 & 125; a “perfect storm for stratification”. Ideally the authors would refit their association models allowing for local-to-*LPA* ancestry in JHS but at minimum I recommend that this “limitation of inference” is noted in the discussion.

Author Response:

The reviewer highlights an important point – as $Lp(a)$ is strongly influenced by the *LPA* locus and by ethnicity, ancestry at the *LPA* locus is likely to be important to consider in association analyses and that lack of consideration may lead to false associations. As the reviewer also highlights, our study used genome-wide estimations of ancestry in association analyses. Using this approach, we observe no systematic deviation of test statistics in genome-wide association analyses compared to the expectation (quantile-quantile plots are presented in Supplementary Figure 11). Furthermore, prior $Lp(a)$ analyses among African Americans (including in JHS) find that in a multivariable model including both genome-wide admixture and *LPA*-locus admixture, genome-wide admixture was fully accounted for by *LPA* local admixture (PMID: 21283670). Our and these prior observations suggest that the genome-wide estimations of ancestry is a suitable proxy for *LPA* locus ancestry in genome-wide association analyses of $Lp(a)$.

Manuscript Amendment:

We have now included in the Discussion:

“Furthermore, given the strong influence of ancestry on $Lp(a)$, adjustment of *LPA* locus ancestry may improve power for genetic association. Indeed, prior analyses of African Americans suggest that genome-wide estimations of ancestry are correlated with *LPA* locus ancestry estimations⁴³.”

2. Line 128 and Suppl Figure 5: A “ $0.3 < \rho < 0.5$ ” is typically referred to as a moderate correlation, the p-value is tiny more due to the huge sample size than the magnitude of ρ .

Manuscript Amendment:

“Among JHS individuals with both $Lp(a)$ and $Lp(a)$ -C available, the concentrations between these phenotypes were moderately correlated (Spearman correlation (R_s) = 0.46, $P = 2.4 \times 10^{-143}$).”

3. Line 144/145: rho (Figure 7c) or rho² (line 145); this I suggest is a “very strong” (magnitude of rho) as well as a “robust” (to outliers) non-parametric correlation.

Manuscript Amendment:

“To evaluate the precision of our KIV2-CN estimates, we utilized 123 pairs of siblings from JHS that were confidently identical-by-descent at both LPA 1Mb window haplotypes (genotype concordance > 99%), and found a very strong and robust correlation between sibling pair KIV2 copy number estimates ($r^2=0.989$) (Supplementary Fig. 7a-d). ”

4. Lines 158 & 159: Fig 2c (not Fig 3b?) shows a strong correlation between genotyped and imputed KIV2-CN, but the regression slope is <1, add a y=x reference line to highlight this please (as I don't think many in the field are aware of this feature of genotype imputation).

Manuscript Amendment:

Figure 2c has been amended adding a y=x reference line as suggested:

5. Line 242: Confirm rho or rho²?

Author Response:

Rho², as specified in the text.

6. Line 305: “10 incident clinical events” means “10 different clinical phenotypes”, not n=10 patients? Rephrase?

Manuscript Amendment:

“Association of GRS+KIV2-CN with 10 incident clinical phenotypes from the FIN imputation dataset (N=27,344) (Fig. 7a, Supplementary Table 30) demonstrated...”

7. Lines 309 – 311: This introduces a key result, with the umbrella “cardiovascular disease” model selected from the 40 results listed in Suppl. Table 30, perhaps as it shows the maximal discrepancy in HR contrasting GRS vs. KIV2-CV? Fig. 7a does though show a

similar, albeit slightly less extreme, difference in HR for the two predictors for the MI or the slightly broader CHD phenotype. These latter two phenotypes are arguably “cleaner” and less pathophysiologically heterogeneous than “cardiovascular disease”, and have been widely used in GWAS with great success. So I suggest that results drawn from Fig. 7a are sufficient to satisfactorily make the point here.

Manuscript Amendment:

“For given effect on Lp(a), the GRS had a larger effect on incident coronary heart disease risk (HR 1.36/Lp(a) SD, $P = 7.6 \times 10^{-8}$) than KIV2-CN (HR 1.03/Lp(a) SD, $P = 0.17$). Similar trends were observed for incident myocardial infarction.”

8. Line 925: I think you mean Suppl Fig 8a (& 8b) (not Suppl Fig 6a). Would be helpful to show the MSE for >61 variants in Suppl. Fig 8b to fully appreciate the stopping decision.

Author Response:

Supplementary Fig. 8a shows the mean squared error (MSE) from the LASSO model across all of the variants. The LASSO model was used to perform variant selection and determine the stopping decision, whereby the use of 61 variants minimized the MSE with the minimum numbers of estimated parameters.

Subsequently, these individual relative importance of each of the 61 variants was estimated using a random forest model as depicted in Supplementary Fig. 8b and Figure 2b. Hence, at this stage, the full model includes 61 variants and additional variants are not included. The additional variants (beyond 61) and resultant MSE are depicted currently in Supplementary Figure 8a.

Manuscript Amendment:

The text accompanying these supplementary figures has been edited to clarify this point.

Reviewer #1 (Remarks to the Author):

The author responded to all previous comments. I have no further remarks

Reviewer #2 had no further comments to the authors